# Mitigating Spurious Correlation via Distributionally Robust Learning with Hierarchical Ambiguity Sets

## Abstract

Conventional supervised learning methods are often vulnerable to spurious correlations, particularly under distribution shifts in test data. To address this issue, several approaches, most notably Group DRO, have been developed. While these methods are highly robust to subpopulation or group shifts, they remain vulnerable to intra-group distributional shifts, which frequently occur in minority groups with limited samples. We propose a hierarchical extension of Group DRO that addresses both inter-group and intra-group uncertainties, providing robustness to distribution shifts at multiple levels. We also introduce new benchmark settings that simulate realistic minority group distribution shifts—an important yet previously underexplored challenge in spurious correlation research. Our method demonstrates strong robustness under these conditions—where existing robust learning methods consistently fail—while also achieving superior performance on standard benchmarks. These results highlight the importance of broadening the ambiguity set to better capture both inter-group and intra-group distributional uncertainties.

## 1 Introduction

In recent years, machine learning methods have achieved remarkable success across a wide range of applications. An important objective of many machine learning methods is to learn model parameters that minimize the population risk, which is the population expectation of the loss function. Given training data and model parameters, the population risk can be approximated by the empirical risk, defined as the sample-averaged loss. Therefore, model parameters can be learned by minimizing the empirical risk, which is known as the empirical risk minimization (ERM) principle.

The underlying assumption of ERM-based methods is that the unseen future data, often referred to as test data, share the same distribution as the training data. However, in many real-world problems, the test data may follow a different distribution from the training data for various reasons. A notable example is subpopulation shift, where the training population consists of several groups (subpopulations), and the proportion of each group in the test data differs from that in the training data [42, 10, 52].

In many instances of subpopulation shifts, the group indicator is spuriously correlated with the target label or response variable. For example, in the widely studied Waterbirds dataset [42], the target label (e.g., "Waterbird" or "Landbird") is spuriously correlated with the background environment (e.g., water or land). As a result, ERM-based models tend to associate "Waterbird" primarily with water backgrounds, leading to significant performance degradation on minority groups, such as waterbirds appearing against land backgrounds. These vulnerabilities extend beyond controlled benchmarks, posing substantial risks in domains such as healthcare [54, 4], fairness [20, 9, 38], and autonomous driving [57].

Over the past few years, a growing body of research has focused on mitigating these spurious correlations to ensure more reliable and stable model performance. Among the proposed methodologies, group distributionally robust optimization (Group DRO) [42] has emerged as a foundational approach. By partitioning the data into predefined groups and optimizing for the worst group loss, Group DRO effectively minimizes the model's reliance on spurious correlations tied to specific subsets of data. This framework has inspired a wide range of subsequent methods, such as JTT [30], SAA [35], PG-DRO [17], DISC [51] and GIC [19], which commonly employ a two-step strategy: first identifying latent groups and then utilizing established robust training approaches—frequently Group DRO itself—to enhance model robustness.

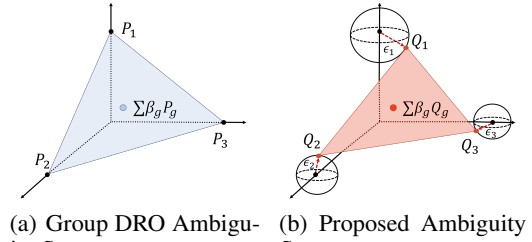

(a) Group DRO Ambiguity Set    (b) Proposed Ambiguity Set

Figure 1: Comparison of the Group DRO ambiguity set (a) and our hierarchical extension (b). While Group DRO restricts uncertainty to mixtures of group distributions, our approach introduces additional within-group uncertainty (indicated by red dashed arrows), offering robustness to both inter-group and intra-group distributional shifts. (For visualization, we assume the 3-dimensional space in the figure represents a probability space, where each point corresponds to a probability distribution.)

While these methods have significantly advanced the field, they primarily focus on minimizing the worst-group loss under the assumption that each group's training distribution reliably represents its true underlying distribution. However, this assumption often fails in practice—especially for minority groups with limited samples—where within-group distributional shifts naturally arise as a consequence of underrepresentation in the training data [7, 13, 15]. This limitation highlights the need for a more flexible and robust approach that accounts for uncertainty not only across groups, but also within them.

In this work, we address these limitations by introducing a hierarchical ambiguity set within the Group DRO framework, capturing both inter-group and intra-group uncertainties. As illustrated in Figure 1, while conventional Group DRO focuses on robustness only to shifts in group proportions by minimizing the worst-group risk, our approach extends this perspective by additionally modeling uncertainty within individual groups.

Technically, we employ a Wasserstein-distance-based formulation, which has recently garnered significant theoretical and empirical support for its efficacy in designing distributionally robust learning methods [49, 28, 8, 5]. By defining a semantically meaningful cost function in a latent space, this formulation flexibly accommodates variations in the underlying data-generating mechanisms within each group. Consequently, our hierarchical ambiguity set enables the model to maintain robustness across a broader spectrum of distributional deviations, particularly for minority groups that are underrepresented in the training data.

Our main contributions are threefold:

- We re-examine Group DRO from a distributionally robust perspective and introduce a novel hierarchical ambiguity set that captures both inter-group and intra-group uncertainties, constituting a fundamental advancement over previous methods that build on Group DRO.

- We establish more realistic and challenging evaluation scenarios by modifying standard benchmarks—Waterbirds, CelebA, and CMNIST—to introduce minority group distribution shifts that prior work has overlooked. This enables more faithful assessment of robustness under real-world minority underrepresentation.

- Through extensive experiments, we show that our hierarchical approach consistently outperforms conventional Group DRO and related robust learning methods, underscoring the practical importance of a more flexible and theoretically grounded extension of the Group DRO framework. To the best of our knowledge, this is the first work to explicitly address intra-group distribution shifts alongside group-level spurious correlations in standard benchmark settings, highlighting a novel contribution to robust learning.

## 2  Related work

Using explicit group labels is a well-established approach to achieving robust performance on underrepresented subpopulations. [42] pioneered partitioning data by known group annotations and minimizing worst-group loss. Subsequent works extend this paradigm in multiple directions: Along similar lines, [27] expands Group DRO's ambiguity set from convex to affine combinations of group distributions, although it still assumes fixed conditional distributions within each group. Meanwhile, [24] rebalances distributions via subsampling, [12] employs a staged expansion of a group-balanced set, and [53] uses Mixup-based augmentations (e.g., CutMix, Manifold Mix) to learn more invariant features. [21] shows that even standard ERM can yield robust feature representations when combined with selective reweighting, and [40] further leverages inter-group interactions to identify shared features that enhance distributional robustness.

In parallel, a growing body of work addresses scenarios where group annotations are unavailable or prohibitively expensive, motivating the need to infer or approximate group structure. Building on [24], [29] extends last-layer retraining to settings with minimal or no group annotations. Some methods employ limited validation sets to discover latent groups and then apply Group DRO [42] for robust optimization [35, 17]. Indeed, the model's loss (or its alternatives) often helps identify underrepresented subpopulations; for example, [34, 30, 41] use high-loss samples to recognize minority groups. Other approaches draw on diverse cues: [1, 11] infer groups by maximizing violations of the invariant risk minimization principle [2], while [3] employs masking to reduce reliance on spurious features. [45, 44] instead cluster feature embeddings to discover latent groups and identify pseudo-attributes for debiasing. Likewise, [56] proposes a two-stage contrastive learning framework by aligning samples with the same class but different spurious attributes, and [51] constructs a "concept bank" of candidate spurious attributes for robust partitioning. Another line of work identifies spurious features by comparing the training set with a carefully selected reference dataset [19] or removes examples that disproportionately degrade worst-group accuracy [22].

Beyond these established techniques, emerging efforts explore scenarios with imperfect group partitions [59] or multiple spurious attributes [39, 23]. These studies challenge the assumption that spurious attributes remain fixed or neatly separated, prompting methods that better accommodate complex, real-world data. Notably, group-inference approaches, such as those proposed by [30] and [41], primarily focus on identifying underrepresented samples. However, these methods pay less attention to how faithfully those samples reflect the underlying distribution. Our work explicitly challenges the assumption that minority-group data reliably mirror their underlying distribution, modeling potential discrepancies within these subpopulations and underscoring the need for frameworks that capture not only which groups matter but also how accurately they represent real-world conditions.

## 3  Preliminaries

**Problem Setup.**  We consider a supervised learning problem where each observation consists of input features $X \in \mathcal{X}$ and a label $Y \in \mathcal{Y}$. Let $\{(x_i, y_i)\}_{i=1}^n$ be the training data and $\Theta$ be the parameter space. Assuming that the test data follows the same distribution as the training data, an important goal of various machine learning methods is to minimize the risk (also referred to as the test or generalization error)

$$\mathbb{E}_P\big[\mathcal{L}(f^\theta(X), Y)\big] \tag{1}$$

over $\Theta$, where $f^\theta$ is a function parametrized by $\theta$, $\mathcal{L}$ is a standard loss function, such as the cross-entropy, and $\mathbb{E}_P$ denotes the expectation under the population distribution $P$. To achieve this, one may solve the following optimization problem:

$$\inf_{\theta \in \Theta} \left\{ \mathbb{E}_{\hat{P}}[\ell(\theta; (X, Y))] = \frac{1}{n} \sum_{i=1}^n \ell\big(\theta; (x_i, y_i)\big) \right\},$$

often referred to as the ERM problem, where $\hat{P}$ is the empirical measure and $\ell(\theta; (X, Y))$ is shorthand for $\mathcal{L}(f^\theta(X), Y)$.

In addition to the above basic setting, we assume that the data are partitioned into multiple groups, with an indicator variable $G \in \mathcal{G}$; thus, the training data can be expressed as $\{(x_i, y_i, g_i)\}_{i=1}^n$. In particular, we assume that this indicator variable is spuriously correlated with the label $Y$. More

specifically, in all our examples, we assume the existence of a spurious attribute $A \in \mathcal{A}$, and that the group variable $G = (Y, A)$ is a pair involving the response variable $Y$. Hence, $\mathcal{G} = \mathcal{Y} \times \mathcal{A}$. With adjusted notation, we write $\mathcal{G} = \{1, \ldots, m\}$.

In the Waterbirds dataset, for example, the task is to classify birds as Waterbird or Landbird, with a spurious attribute being the background type (Water Background or Land Background), which results in four distinct groups. This dataset provides a classic example of spurious correlation: most waterbirds ($Y =$ Waterbird) are found against water backgrounds ($A =$ Water Background), leading models to rely excessively on background features. This overreliance substantially diminishes the performance of ERM on underrepresented groups, such as waterbirds on land backgrounds.

**Group DRO.** Group DRO [42] was devised to address the aforementioned issue caused by the spurious attribute. In the Group DRO, the training data are modeled as instances from a mixture distribution $P = \sum_{g=1}^{m} \alpha_g P_g$, where $P_g$ denotes the conditional distribution of $(X, Y)$ given $G = g$, and $\alpha = (\alpha_1, \ldots, \alpha_m)$ represents the mixing proportion. Thus, each group forms a subpopulation within the training data. Instead of minimizing the population risk (1), Group DRO aims to minimize

$$\inf_{\theta \in \Theta} \max_{g \in \mathcal{G}} \mathbb{E}_{P_g}\big[\ell(\theta; (X, Y))\big], \tag{2}$$

which corresponds to the risk of the worst-performing group. Consequently, this procedure is highly robust to the subpopulation shift described in the introduction.

The Group DRO formulation (2) involves optimization over the discrete group variable $g$, posing a computational challenge for practical use. [42] showed that the problem can be reformulated into an equivalent form

$$\inf_{\theta \in \Theta} \sup_{\beta \in \Delta_{m-1}} \sum_{g=1}^{m} \beta_g \mathbb{E}_{P_g}\big[\ell(\theta; (X, Y))\big],$$

that involves continuous variables only, where $\Delta_{m-1} = \{\beta : \beta_g \geq 0, \sum_{g=1}^{m} \beta_g = 1\}$ is the $(m-1)$-simplex.

Group DRO can be understood as an instance of standard DRO [6, 14] with a specific ambiguity set. Note that the standard DRO formulation is given as

$$\inf_{\theta \in \Theta} \sup_{Q \in \mathcal{Q}} \mathbb{E}_Q\big[\ell(\theta; (X, Y))\big], \tag{3}$$

where $\mathcal{Q}$ is a class of distributions, commonly referred to as the *ambiguity (or uncertainty) set*. The Group DRO formulation (2) is a specific case of DRO (3), with

$$\mathcal{Q} := \left\{ \sum_{g=1}^{m} \beta_g P_g \ : \ \beta \in \Delta_{m-1} \right\}. \tag{4}$$

Note that in frequently used DRO frameworks, the ambiguity set $\mathcal{Q}$ is often defined as a small neighborhood with respect to a standard (pseudo-)metric, such as the Wasserstein distance [33, 16] and $f$-divergence [36, 32].

# 4 Proposed Method

## 4.1 Hierarchical Ambiguity Sets

In this section, we propose a hierarchical extension of Group DRO to be robust to distribution shifts at multiple levels. The proposed method is devised to capture both inter-group and intra-group uncertainties in modeling the distributional shifts.

**High-level Formulation.** As in Group DRO, we model the training distribution as a mixture of the form $P = \sum_{g=1}^{m} \alpha_g P_g$. To model the distributional uncertainty, we consider the DRO formulation (3) with a *hierarchical ambiguity set* $\mathcal{Q}$, defined as

$$\mathcal{Q} = \left\{ \sum_{g=1}^{m} \beta_g Q_g \ : \ \begin{array}{l} \beta \in \Delta_{m-1}, \ d_1(\beta, \alpha) \leq \rho, \\ d_2(Q_g, P_g) \leq \epsilon_g \quad \forall g \end{array} \right\}, \tag{5}$$

where $d_1$ and $d_2$ are suitable metrics on $\Delta_{m-1}$ and the class of distributions for $(X, Y)$, respectively, and $\rho, \epsilon_g > 0$ are radii that determine the size of the ambiguity set.

The ambiguity set $\mathcal{Q}$ has a two-level hierarchy. The first level is controlled by the mixing proportion $\beta$. It accounts for uncertainty in the proportion of each subpopulation or group. Such uncertainty can arise, for example, if certain minority groups appear more frequently in evaluation settings than in the training set, thereby increasing their probability of occurrence and potentially amplifying spurious correlations if not properly addressed [47]. At the second level, the distributional shift in each group is considered to capture within-group variability.

By jointly accounting for changes in the group proportion $\alpha$ and the conditional distributions $\{P_g\}_{g=1}^m$, the proposed framework provides two levels of robustness: inter-group generalization and resilience to intra-group variability. This dual modeling of real-world uncertainties enables the proposed method to address a broader range of distributional shifts compared to Group DRO (2) alone or standard DRO (3), which uses a standard (pseudo-)metric neighborhood as its ambiguity set.

**Relationship to Group DRO and Standard DRO.**  The ambiguity set (4) used in Group DRO is a special case of the proposed ambiguity set (5). In particular, (4) can be obtained by setting $\rho = \infty$ and $\epsilon_g = 0$.

While standard DRO, which uses a standard metric neighborhood as an ambiguity set, can also be understood as a special case of the proposed method, the philosophy of the proposed ambiguity set differs from that of the standard ones. In standard DRO, the ambiguity set is taken as a small neighborhood with respect to a standard (pseudo-)metric. In contrast, we allow a large value for $\rho$, the radius that determines robustness to group proportions. Hence, distributions that are far from $P$ with respect to standard metrics can also belong to the ambiguity set (5).

**Detailed Formulation.**  The choice of $d_1$ is not critical because most reasonable metrics are equivalent. In the remainder of this paper, we set $\rho = \infty$.

For $d_2$, we consider the infinite-order Wasserstein distance for computational convenience. Recall the definition of the Wasserstein distance of order $p \in [1, \infty)$:

$$W_p(Q, P) = \inf_\gamma \left\{ \left( \int c\big((x, y), (x', y')\big)^p \, d\gamma \right)^{\frac{1}{p}} \right\},$$

where the infimum is taken over every coupling $\gamma$ of $Q$ and $P$, and $c(\cdot, \cdot)$ is a cost function. The infinite-order Wasserstein distance is defined as $W_\infty(Q, P) = \sup_{p \geq 1} W_p(Q, P)$, with a variational representation

$$W_\infty(Q, P) = \inf \left\{ \epsilon > 0 \ : \ \begin{array}{c} Q(A) \leq P(A^\epsilon) \\ \text{for every Borel set } A \end{array} \right\}, \tag{6}$$

where $A^\epsilon$ denotes the $\epsilon$-enlargement of $A$; see [18].

The cost function is defined in a latent semantic space, which is more effective than defining it in the space of raw data [55, 26]. Specifically, we employ a deep neural network $f^\theta$ of depth $L$, defined as

$$f^\theta(x) = f_L^\theta \left( f_{L-1}^\theta \left( \dots f_1^\theta(x) \right) \right),$$

and take the output of the $(L-1)$-th layer (before the final fully connected layer) as the semantic representation:

$$z(x) := f_{L-1}^\theta \left( f_{L-2}^\theta \left( \dots f_1^\theta(x) \right) \right). \tag{7}$$

We then define the cost function $c(\cdot, \cdot)$ as

$$c\big((x, y), (x', y')\big) \ = \ \begin{cases} \|z(x) - z(x')\|, & \text{if } y = y', \\ \infty, & \text{otherwise.} \end{cases}$$

Note that under our definition, $W_p(P, Q) = \infty$ if the marginals $P$ and $Q$ of $Y$ differ. In all our applications, the group indicator $G$ is defined as a pair $(Y, A)$; hence, this definition does not cause any issues.

**Proposed Hierarchical DRO Formulation.** To sum up, the proposed hierarchical DRO can be written in the standard form (3) with the ambiguity set

$$\mathcal{Q} = \left\{ \sum_{g=1}^{m} \beta_g Q_g \ : \ \begin{array}{l} \beta \in \Delta_{m-1}, \\ W_\infty(Q_g, P_g) \le \epsilon_g \quad \forall g \end{array} \right\}. \tag{8}$$

The flexibility in $\beta \in \Delta_{m-1}$ allows the group proportion to differ from $\alpha$, which enables adaptation to new or changing subpopulation frequencies without introducing entirely new groups. With the constraint $W_\infty(Q_g, P_g) \le \epsilon_g$, we accommodate plausible instance-level shifts within each group. Leveraging the semantic cost $c(\cdot, \cdot)$ allows the model to capture meaningful perturbations in high-dimensional feature spaces without conflating different class labels.

## 4.2 Algorithm

In this subsection, we provide an algorithm to solve the proposed DRO with the ambiguity set (8). We set $\epsilon_g = \epsilon/\sqrt{n_g}$, where $n_g$ is the size of group $g$ in the training data, and $\epsilon$ is a tunable hyperparameter that controls the degree of robustness to within-group distributional shifts. Intuitively, the fewer samples a group has, the more cautiously its potential distributional variations must be accounted for. See Section 4.3 for details on the selection of the tuning parameter $\epsilon$.

Due to the hierarchical structure of the ambiguity set in (8), solving the resulting DRO problem is not straightforward. We therefore begin by reformulating the hierarchical DRO into a tractable optimization problem. We formally state the resulting formulation in the following theorem. The proof is provided in Appendix A.

**Theorem 4.1.** *Let $\mathcal{Q}$ be the ambiguity set defined in* (8). *Then, the corresponding distributionally robust optimization problem*

$$\inf_{\theta \in \Theta} \sup_{Q \in \mathcal{Q}} \mathbb{E}_Q[\ell(\theta; (X, Y))]$$

*is upper-bounded by the following surrogate objective:*

$$\inf_{\theta \in \Theta} \sup_{\beta \in \Delta_{m-1}} \sum_{g=1}^{m} \beta_g \mathbb{E}_{P_g} \left[ \sup_{z' : \|z' - z(X)\| \le \epsilon_g} \mathcal{L}\big(f_L^\theta(z'), Y\big) \right]. \tag{9}$$

Intuitively, Theorem 4.1 shows that the worst-case risk over our hierarchical ambiguity set can be conservatively over-approximated by an adversarial perturbation problem in the latent space, where the inner maximization is weighted by the worst-case group proportions $\beta$. We therefore minimize the surrogate objective (9) via a coordinate-wise procedure, as detailed next.

**Proposed Iterative Training Procedure.** For a given $\theta$, let $z_i'$ denote the maximizer of the map

$$z' \mapsto \mathcal{L}\big(f_L^\theta(z'), y_i\big)$$

over the set $\{z' : \|z' - z(x_i)\| \le \epsilon_{g_i}\}$. To solve the optimization problem (9), we iteratively update $\beta$, $\theta$ and semantic variables $z_i'$ coordinate-wise as below. A pseudo-code for a minibatch size of 1 is provided in Algorithm 1.

1. *Update of $z'$.* For given $\theta$, $z_i'$ can be approximated by one-step projected gradient ascent, ensuring that $\|z' - z(x_i)\| \le \epsilon_{g_i}$. (Lines 6–8)

2. *Update of $\beta$.* For given $\theta$ and $z_i'$, $\beta$ can be computed using exponentiated gradient ascent, a variant of mirror descent with negative Shannon entropy [42]. (Lines 10–12)

3. *Update of $\theta$.* For a given $\beta$ and $z_i'$, we update $\theta$ using stochastic gradient descent. (Line 13)

A convergence guarantee under convexity assumptions is established in Appendix B, showing that the algorithm achieves an $O(1/\sqrt{T})$ convergence rate.

## 4.3 Selection of $\epsilon$

As is common in DRO problems, the selection of the size of an ambiguity set, $\epsilon$ in our problem, is a challenging task. To address this challenge, we propose a heuristic data-driven procedure that is

**Algorithm 1** DRO with a Hierarchical Ambiguity Set

---

1: **Input:** Step sizes $\eta_\beta, \eta_\theta, \eta_z$; initial parameters $\theta^{(0)}, \beta^{(0)}$; number of iterations $T$
2: **for** $t = 1$ **to** $T$ **do**
3:     Sample $g \sim \text{Uniform}(1, \ldots, m)$
4:     Sample $(x, y) \sim P_g$
5:     Initialize $z' \leftarrow z(x)$
6:     $z' \leftarrow z' + \eta_z \nabla_{z'} \mathcal{L}\big(f_L^{\theta^{(t-1)}}(z'), y\big)$
7:     **if** $\|z' - z(x)\| > \epsilon_g$ **then**
8:         $z' \leftarrow \text{Proj}_{\|z'-z(x)\| \leq \epsilon_g}(z')$
9:     **end if**
10:    Update $\beta' \leftarrow \beta^{(t-1)}$
11:    Update $\beta'_g \leftarrow \beta'_g \exp\Big(\eta_\beta \mathcal{L}\big(f_L^{\theta^{(t-1)}}(z'), y\big)\Big)$
12:    Normalize $\beta^{(t)} \leftarrow \beta' / \sum \beta'_{g'}$
13:    Update $\theta^{(t)} \leftarrow \theta^{(t-1)} - \eta_\theta \beta_g^{(t)} \nabla_\theta \mathcal{L}\big(f_L^{\theta^{(t-1)}}(z'), y\big)$
14: **end for**

---

similar to cross-validation, but partitions the training data based on the order of a one-dimensional t-SNE [48] feature. Similar procedures have been considered in the literature [15].

Specifically, we project each $z(x_i)$ onto a one-dimensional space using t-SNE. This allows us to rank samples within each group and split them into five quantiles. We focus on two extreme quantiles (the top 20% and bottom 20%), each held out as a validation set in turn, with the remaining 80% used for training. By training and evaluating on these opposite extremes, we simulate realistic distribution shifts that disproportionately affect minority groups.

We then measure the model's performance under both setups and select the value of $\epsilon$ that maximizes minority-group accuracy on average. This ensures that the chosen perturbation radius is robust to distributional shifts and provides meaningful protection for underrepresented subpopulations in practice.

## 5 Experiments

### 5.1 Dataset

We conduct experiments on three widely used benchmark datasets, CMNIST, Waterbirds, and CelebA, each exhibiting known spurious correlations between the label and an irrelevant attribute. All datasets include a minority group that is underrepresented, rendering them susceptible to distributional shifts.

**Original Datasets.**

- **CMNIST** [2]: A colored variant of MNIST, split into four groups based on digit label (*digits 0–4 as label 0*, and *digits 5–9 as label 1*) and color (*red* vs. *green*). The color is spuriously correlated with the digit label in the training set.
- **Waterbirds** [42]: Created by combining bird images from CUB [50] with backgrounds from Places [58], yielding four groups based on *(bird type, background)*. The minority group *(waterbird, land background)* typically has few samples.
- **CelebA** [31]: A facial attribute dataset used here for classifying *blond* vs. *non-blond* hair, where *gender* acts as a spurious attribute. The minority group *(blond hair, male)* is significantly underrepresented.

**Modified Datasets with Minority Group Shifts.** To rigorously test our approach under more realistic distribution shifts, we construct modified versions of the above datasets by inducing intra-group shifts specifically in each minority group:

- **Shifted CMNIST**: Rotate all images in the minority group *(label 1, red)* by $90°$ at test time, while keeping them unrotated at training time.
- **Shifted Waterbirds**: Restrict the training set's minority group *(waterbird, land background)* to only *waterfowls*, and the test set's minority group to only *seabirds*.

Table 1: Worst-group and average accuracy on CMNIST, Waterbirds, and CelebA under shifted distributions. All results are averaged over three runs with different random seeds. Boldface indicates the best performance, while underlined numbers denote the second-best.

| Method | Group label | Shifted CMNIST | | Shifted Waterbirds | | Shifted CelebA | |
|---|---|---|---|---|---|---|---|
| | | Worst Acc | Average Acc | Worst Acc | Average Acc | Worst Acc | Average Acc |
| GroupDRO | ✓ | 65.9±8.2 | 74.0±0.7 | 91.7±0.3 | 94.9±0.1 | 59.8±3.2 | 92.4±0.3 |
| LISA | ✓ | 42.9±10.0 | 59.8±4.5 | 79.1±1.8 | 94.2±0.3 | 60.6±1.1 | 92.1±0.2 |
| DFR$^{tr}$ | ✓ | 28.0±4.9 | 47.8±1.9 | 89.2±1.5 | 96.3±0.4 | 50.3±3.5 | 90.5±0.4 |
| PDE | ✓ | 65.3±11.1 | 71.3±6.2 | 84.4±4.6 | 92.0±0.6 | 56.3±11.2 | 91.6±0.4 |
| Ours | ✓ | **71.8**±2.8 | 75.0±0.4 | **93.7**±0.2 | 94.6±0.1 | **72.1**±2.0 | 91.3±0.1 |

Table 2: Worst-group and average accuracy on CMNIST, Waterbirds, and CelebA under their original (unshifted) distributions.

| Method | Group label | CMNIST | | Waterbirds | | CelebA | |
|---|---|---|---|---|---|---|---|
| | | Worst Acc | Average Acc | Worst Acc | Average Acc | Worst Acc | Average Acc |
| ERM | ✗ | 3.4±0.9 | 12.9±0.8 | 62.6±0.3 | 97.3±1.0 | 47.7±2.1 | 94.9±0.3 |
| JTT | ✗ | 67.3±5.1 | 76.4±3.3 | 83.8±1.2 | 89.3±0.7 | 81.5±1.7 | 88.1±0.3 |
| CnC | ✗ | – | – | 88.5±0.3 | 90.9±0.1 | 88.8±0.9 | 89.9±0.5 |
| GIC | ✗ | 72.2±0.5 | 73.2±0.2 | 86.3±0.1 | 89.6±1.3 | 89.4±0.2 | 91.9±0.1 |
| SSA | ✗ | 71.1±0.4 | 75.0±0.3 | 89.0±0.6 | 92.2±0.9 | 89.8±1.3 | 92.8±0.1 |
| GroupDRO | ✓ | 73.1±0.3 | 74.8±0.2 | 90.6±0.2 | 92.7±0.1 | 89.3±1.3 | 92.6±0.3 |
| LISA | ✓ | 73.3±0.2 | 74.0±0.1 | 89.2±0.6 | 91.8±0.3 | 89.3±1.1 | 92.4±0.4 |
| DFR$^{tr}$ | ✓ | 59.8±0.4 | 62.1±0.2 | 90.2±0.8 | 97.0±0.3 | 80.7±2.4 | 90.6±0.7 |
| PDE | ✓ | 72.6±0.7 | 73.0±0.4 | 90.3±0.3 | 92.4±0.8 | **91.0**±0.4 | 92.0±0.6 |
| Ours | ✓ | **73.6**±0.3 | 75.1±0.5 | **90.8**±0.2 | 92.6±0.2 | 90.4±0.3 | 92.7±0.0 |

- **Shifted CelebA**: For minority group *(blond hair, male)*, include only *no-glasses* images in training and only *with-glasses* images at test time.

These modifications reflect real-world scenarios where underrepresented groups not only appear more frequently but also exhibit subtle changes. Further details and illustrative examples are provided in Appendix D.

## 5.2 Baselines

We compare our method to several representative baselines: ERM, Group DRO [42], JTT [30], CnC [56], SAA [35], LISA [53], DFR [24], PDE [12], and GIC [19]. These methods range from direct robust learning (e.g., Group DRO) to two-step pipelines that first infer group membership and then apply robust training (e.g., SAA, GIC). Detailed descriptions are provided in Appendix E.

For our newly constructed datasets incorporating minority-group distribution shifts, we conducted experiments focusing on Group DRO, LISA, DFR, and PDE. Unlike methods that infer group labels and then rely on a separate robust training step, these four baselines—like our proposed approach—directly utilize known group information. This distinction provides a more consistent and fair comparison in scenarios where explicit group labels are available.

## 5.3 Evaluation

**Metrics.** We consider two metrics: *worst-group accuracy* and *average accuracy*. The worst-group accuracy is obtained by evaluating accuracy on each group and taking the minimum across all groups, providing insight into how a method performs if the test distribution is heavily skewed toward the most challenging subgroup. Meanwhile, the average accuracy is computed as the weighted average of group accuracy, where the weights are proportional to the group sizes in the training data, reflecting overall performance but offering less visibility into group-specific disparities.

**Model Selection.** Following [42] and related methods, we select hyperparameters and stopping criteria based on the highest worst-group validation accuracy. In particular, for scenarios involving minority group shifts, we adopt the data-driven tuning procedure from Section 4.3 to determine the perturbation parameter $\epsilon$.

## 5.4 Results

**Performance on Shifted Distributions.**
Under shifted distributions (Table 1), our method demonstrates clear superiority in worst-group accuracy across all three benchmarks (CMNIST, Waterbirds, and CelebA). On CMNIST, LISA and DFR degrade substantially, highlighting their vulnerability to intra-group shifts. By contrast, our framework maintains high worst-group accuracy.

Table 3: Worst-group and average accuracy on Waterbirds under minority group shifted distributions and Corrected Waterbirds on the original dataset with corrected labels.

| Method | Shifted Waterbirds | | Corrected Waterbirds | |
|--------|-----------|---------|-----------|---------|
| | Worst Acc | Avg Acc | Worst Acc | Avg Acc |
| Group DRO | 91.7±0.3 | 94.9±0.1 | 94.1±0.6 | 94.7±0.0 |
| Ours | **93.7**±0.2 | 94.6±0.1 | **95.1**±0.4 | 96.3±0.0 |

For Waterbirds, which involves a moderate shift in species composition within the minority group, most baselines experience notable drops in worst-group accuracy. In contrast, our approach maintains robust worst-group accuracy, indicating its capacity to adapt to intra-group variability. Interestingly, both Group DRO and our method report higher worst-group accuracy in the shifted case than in the original Waterbirds dataset (Table 2); however, this discrepancy arises from a known mislabeling issue [3], where three bird species labeled as "waterbird" should actually be "landbird." To verify this, we correct the mislabeled samples in the original dataset and report results in Table 3: under the corrected labels, both Group DRO and our method exhibit the expected pattern, performing better in the unshifted setting than under the minority-group shift. Notably, our approach outperforms Group DRO in both scenarios, confirming its robustness even after label corrections.

On the more challenging CelebA benchmark, our advantage grows more pronounced. While PDE shows slightly higher worst-group accuracy on the original dataset (Table 2), its performance drops sharply (by about 34.7%) when the minority-group distribution is shifted. These observations underscore the importance of modeling both inter-group and intra-group uncertainties—especially given that minority groups in Waterbirds and CelebA constitute only about 1% of the data and thus are more susceptible to distributional changes. Furthermore, our results highlight that relying on pre-defined test splits with uniformly distributed attributes may offer an overly optimistic view of real-world robustness.

**Performance on Original Distributions.** On the original (unshifted) versions of CMNIST, Waterbirds, and CelebA (Table 2), our method consistently achieves top-tier worst-group accuracy. It secures the highest or near-highest scores across all three benchmarks, confirming that the proposed framework not only excels under distributional shifts but also remains effective when intra-group distributions are stable. Notably, even in these unshifted settings, methods such as Group DRO rely on the strong assumption that each group's training distribution remains valid at test time. As our results show, explicitly modeling distributional uncertainty within minority groups can yield more reliable robustness, highlighting the limitations of approaches that treat group distributions as fixed. By addressing potential discrepancies at both the inter-group and intra-group levels, our framework provides a stronger foundation for real-world applications.

## 6 Conclusion

We introduced a distributionally robust optimization framework with a hierarchical ambiguity set that explicitly models both inter-group and intra-group distribution shifts—an often overlooked yet practically crucial scenario for underrepresented subpopulations. We find that even small, realistic shifts in how minority group samples are split between training and testing—without altering group definitions—can result in significant degradation of performance in existing robust methods. In contrast, our approach maintains strong performance by modeling latent variability within each group, offering a theoretically grounded and robust foundation for real-world deployment.

**Limitations and Future Work.** Our experiments focus on image datasets with clear feature labels (e.g., species type, presence of glasses) that enable controlled intra-group shifts. Extending the method to other modalities such as text, where such labels are less accessible, is a promising direction. In addition, the radius parameter $\epsilon$ is chosen heuristically; a more principled or automated selection method is worth investigating. Finally, since real-world data may involve multiple spurious features, extending the framework to multi-spurious settings is a promising future direction.

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

# A  Proof of Theorem 4.1

*Proof.* We begin with a lemma adapted from [46], with minor adjustments to match our framework. This lemma provides an equivalent form for the inner supremum problem of DRO with a $W_\infty$-neighborhood, which is closely related to the representation (6) of $W_\infty$.

**Lemma A.1.** *[46, Proposition 3.1] Let $\theta$ be fixed model parameters, and let $c(\cdot, \cdot)$ be a metric on the input space $\mathcal{X}$. For any distribution $P$ on $\mathcal{X} \times \mathcal{Y}$ and for any $\epsilon \geq 0$,*

$$
\mathbb{E}_P\left[ \sup_{(x,y) \in B_\epsilon(X,Y)} \ell(\theta; (x,y)) \right] = \sup_{W_\infty(Q,P) \leq \epsilon} \mathbb{E}_Q[\ell(\theta; (X,Y))].
$$

*where $B_\epsilon(x,y) = \{(x',y') : c((x,y),(x',y')) \leq \epsilon\}$.*

With the ambiguity set (8), the DRO (3) is equivalent to

$$
\inf_{\theta \in \Theta} \left\{ \sup_{\substack{\beta \in \Delta_{m-1} \\ g=1,\dots,m}} \sup_{W_\infty(Q_g, P_g) \leq \epsilon_g} \mathbb{E}_Q[\ell(\theta; (X,Y))] \right\}, \tag{10}
$$

where $Q = \sum_{g=1}^m \beta_g Q_g$.

For a fixed $\theta$, the double supremum in (10) can equivalently be written as

$$
\sup_{\substack{\beta \in \Delta_{m-1} \\ g=1,\dots,m}} \sup_{W_\infty(Q_g, P_g) \leq \epsilon_g} \sum_{g=1}^m \beta_g \, \mathbb{E}_{Q_g}[\ell(\theta; (X,Y))]
$$

$$
= \sup_{\beta \in \Delta_{m-1}} \sum_{g=1}^m \beta_g \sup_{W_\infty(Q_g, P_g) \leq \epsilon_g} \mathbb{E}_{Q_g}[\ell(\theta; (X,Y))].
$$

By applying Lemma A.1, one can upper-bound the inner supremum in the previous display as

$$
\mathbb{E}_{P_g}\left[ \sup_{x : \|z(x) - z(X)\| \leq \epsilon_g} \ell(\theta; (x, Y)) \right] = \mathbb{E}_{P_g}\left[ \sup_{x : \|z(x) - z(X)\| \leq \epsilon_g} \mathcal{L}\big(f_L^\theta(z(x)), Y\big) \right]
$$

$$
\leq \mathbb{E}_{P_g}\left[ \sup_{z' : \|z' - z(X)\| \leq \epsilon_g} \mathcal{L}\big(f_L^\theta(z'), Y\big) \right],
$$

where $x \mapsto z(x)$ denotes the feature map defined in (7). Thus, the original optimization problem is upper-bounded by

$$
\inf_{\theta \in \Theta} \sup_{\beta \in \Delta_{m-1}} \sum_{g=1}^m \beta_g \mathbb{E}_{P_g}\left[ \sup_{z' : \|z' - z(X)\| \leq \epsilon_g} \mathcal{L}\big(f_L^\theta(z'), Y\big) \right],
$$

and completes the proof. $\qquad\qquad\qquad\qquad\qquad\qquad\qquad\qquad\qquad\qquad\qquad\qquad\qquad$ $\square$

# B  Convergence Analysis of Algorithm 1

We analyze convergence via $\varepsilon_T$ of the average iterate $\overline{\theta}^{(1:T)}$ :

$$
\varepsilon_T = \max_{\beta \in \Delta_{m-1}} L\big(\overline{\theta}^{(1:T)}, \beta\big) - \min_{\theta \in \Theta} \max_{\beta \in \Delta_{m-1}} L(\theta, \beta),
$$

where $L(\theta, \beta) := \sum_{g=1}^m \beta_g \mathbb{E}_{P_g}\left[ \sup_{z' : \|z' - z(X)\| \leq \epsilon_g} \mathcal{L}\big(f_L^\theta(z'), Y\big) \right]$. In the convex setting, our method achieves $O(1/\sqrt{T})$.

**Proposition B.1** (Convergence of Algorithm 1). *Suppose $\mathcal{L}\big(f_L^\theta(z),\, y\big)$ is non-negative, convex in $\theta$, $B_\nabla$-Lipschitz in $\theta$, and bounded by $B_\ell$ for all $(x, y)$ in $\mathcal{X} \times \mathcal{Y}$. In addition, let $\|\theta\|_2 \le B_\Theta$ for all $\theta$ in some convex set $\Theta \subset \mathbb{R}^d$, and assume the feature map $z(x)$ is fixed w.r.t. $\theta$. Then, the average iterate of Algorithm 1 achieves an expected error at the rate*

$$\mathbb{E}\big[\varepsilon_T\big] \;\le\; 2\,m\,\sqrt{\frac{10\left(B_\Theta^2\, B_\nabla^2 \;+\; B_\ell^2\, \log m\right)}{T}}.$$

*Proof.* Each iteration samples $G \sim \mathrm{Unif}\{1, \ldots, m\}$ and $(X, Y) \sim P_G$. The resulting joint sample $\xi = (X, Y, G)$ is drawn i.i.d. from the mixture distribution $q := \frac{1}{m}\sum_{g=1}^m P_g$.

For each group $g \in \{1, \ldots, m\}$, define the stochastic loss function

$$F_g(\theta; \xi) := m \cdot \mathbf{1}[G = g] \cdot \sup_{\|z' - z(X)\| \le \epsilon_g} \mathcal{L}\big(f_L^\theta(z'), Y\big),$$

and let

$$f_g(\theta) := \mathbb{E}_{P_g}\left[\sup_{\|z' - z(X)\| \le \epsilon_g} \mathcal{L}\big(f_L^\theta(z'), Y\big)\right].$$

The total objective is then $L(\theta, \beta) = \sum_{g=1}^m \beta_g f_g(\theta)$.

We now verify the conditions required to apply the standard online mirror descent (OMD) regret bound [37]:

(A) Convexity. For each $g$, the inner function $\mathcal{L}(f_L^\theta(z'), Y)$ is convex and non-negative in $\theta$, and the supremum preserves convexity via Danskin's theorem. Thus, $f_g(\theta)$ is convex.

(B) Expectation form. We have

$$\mathbb{E}_{\xi \sim q}\big[F_g(\theta; \xi)\big] = \frac{1}{m}\sum_{g'=1}^m \mathbb{E}_{(X,Y)\sim P_{g'}}\left[m \cdot \mathbf{1}[g' = g] \cdot \sup_{\|z' - z(X)\| \le \epsilon_g} \mathcal{L}\big(f_L^\theta(z'), Y\big)\right] = f_g(\theta).$$

(C) Unbiased subgradients. By Danskin's theorem, the mapping $\theta \mapsto \sup_{z'} \mathcal{L}(f_L^\theta(z'), Y)$ is subdifferentiable. Hence, $\nabla_\theta F_g(\theta; \xi)$ is an unbiased subgradient:

$$\mathbb{E}_{\xi \sim q}\big[\nabla_\theta F_g(\theta; \xi)\big] = \nabla_\theta f_g(\theta).$$

With the conditions (A)–(C) established, and using the boundedness assumptions:

$$\|\theta\|_2 \le B_\Theta, \quad \|\nabla_\theta \mathcal{L}\| \le B_\nabla, \quad \mathcal{L} \le B_\ell,$$

the standard OMD regret bound [37, 42] yields

$$\mathbb{E}\big[\varepsilon_T\big] \;\le\; 2m\,\sqrt{\frac{10\big(B_\Theta^2 B_\nabla^2 + B_\ell^2 \log m\big)}{T}},$$

completing the proof. $\qquad\square$

# C   Interpreting Latent Perturbation Regularization

To clarify the intuition behind our latent perturbation framework, we employ a first-order Taylor expansion of the loss function. This approximation shows that the innermost supremum in our optimization problem can be interpreted as the original loss $\mathcal{L}(f^\theta(x), y)$ plus an additional regularization term involving the dual norm of the gradient with respect to the latent representation. Specifically,

$$\sup_{\|z'-z(x)\|\leq\epsilon}\mathcal{L}(f_L^\theta(z'),y) = \sup_{\|z'-z(x)\|\leq\epsilon}\mathcal{L}\left(f_L^\theta(z(x)+(z'-z(x))),y\right)$$

$$\approx \sup_{\|z'-z(x)\|\leq\epsilon}\left[\mathcal{L}\left(f_L^\theta(z(x)),y\right)+\nabla_z\mathcal{L}\left(f_L^\theta(z(x)),y\right)^\top(z'-z(x))\right]$$

$$= \mathcal{L}\left(f^\theta(x),y\right)+\sup_{\|z'-z(x)\|\leq\epsilon}\nabla_z\mathcal{L}\left(f_L^\theta(z(x)),y\right)^\top(z'-z(x))$$

$$= \mathcal{L}\left(f^\theta(x),y\right)+\epsilon\left\|\nabla_z\mathcal{L}\left(f_L^\theta(z(x)),y\right)\right\|_*,$$

where $\|\cdot\|_*$ denotes the dual norm corresponding to $\|\cdot\|$. This added regularization term penalizes large gradients in the latent space, promoting robustness by ensuring that small perturbations in $z(x)$ do not lead to significant changes in the loss. By minimizing this term alongside the original loss, the model gains stability and improved performance under real-world distributional shifts.

## D  Dataset Details

### D.1  Original Dataset

**Colored MNIST (CMNIST)** [2]. The CMNIST dataset is designed for a noisy digit recognition task, incorporating color as a spurious attribute. The dataset is divided into four distinct groups based on class and color: $g_1 = \{0, \text{green}\}$, $g_2 = \{1, \text{green}\}$, $g_3 = \{0, \text{red}\}$, and $g_4 = \{1, \text{red}\}$. It involves two classes: class 0 includes the digits $(0, 1, 2, 3, 4)$ and class 1 includes the digits $(5, 6, 7, 8, 9)$. The training set consists of 30,000 samples, where for class 0, the ratio of red to green samples is $8 : 2$, while for class 1, this ratio is $2 : 8$. The validation set, which comprises 10,000 samples, maintains an equal distribution of color across both classes, with a $1 : 1$ ratio of red to green samples for each class. The test set includes 20,000 samples and introduces a more pronounced group distribution shift: class 0 has a red to green sample ratio of $1 : 9$, and class 1 has a ratio of $9 : 1$. Following the approach proposed by [2], labels in the dataset are flipped with a probability of 0.25.

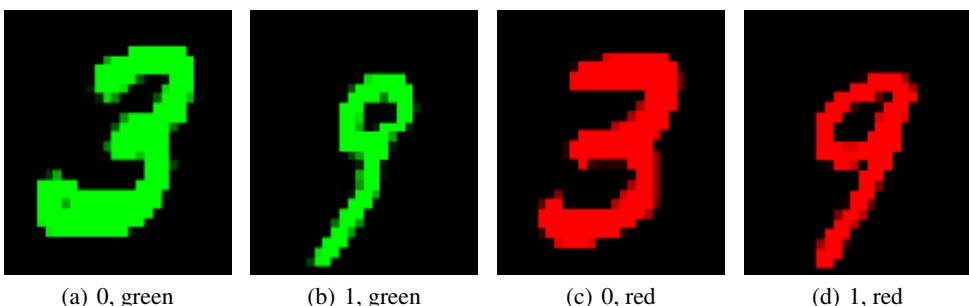

| (a) 0, green | (b) 1, green | (c) 0, red | (d) 1, red |

Figure 2: Example images from the CMNIST dataset. The groups are $g_1 = \{0, \text{green}\}$, $g_2 = \{1, \text{green}\}$, $g_3 = \{0, \text{red}\}$, and $g_4 = \{1, \text{red}\}$.

**Waterbirds** [42]. The Waterbirds dataset is designed to classify images of birds into two categories: "waterbirds" and "landbirds", with a deliberate introduction of spurious correlations between the bird type and the background. The dataset is divided into four distinct groups based on bird type and background: $g_1 = \{\text{landbird, land}\}$, $g_2 = \{\text{landbird, water}\}$, $g_3 = \{\text{waterbird, land}\}$, and $g_4 = \{\text{waterbird, water}\}$. This synthetic dataset is created by combining bird images from the Caltech-UCSD Birds 200-2011 (CUB) dataset [50] with backgrounds from the Places dataset [58]. Waterbird species, such as albatross, auklet, cormorant, frigatebird, and others, are grouped together, while all other species are classified as landbirds. The dataset comprises 4,795 training samples distributed as follows: 3,498 landbirds on land backgrounds, 1,057 waterbirds on water backgrounds, 184 landbirds on water backgrounds, and 56 waterbirds on land backgrounds. This setup highlights the minority groups and the inherent spurious correlations. In contrast to the training set, the validation and test sets are constructed to have an equal number of samples for each group within each class. The minority group, waterbirds on land, emphasizes the skewed distribution of the dataset, making

it suitable for studying the impact of spurious correlations on model performance. The Waterbirds dataset is accessible through the Wilds library in PyTorch [25].

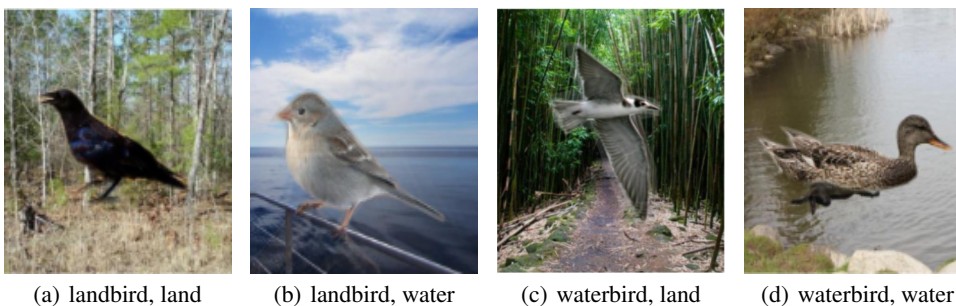

| (a) landbird, land | (b) landbird, water | (c) waterbird, land | (d) waterbird, water |

Figure 3: Example images from the Waterbirds dataset. The groups are $g_1 = \{$landbird, land$\}$, $g_2 = \{$landbird, water$\}$, $g_3 = \{$waterbird, land$\}$, and $g_4 = \{$waterbird, water$\}$.

**CelebA** [31]. The CelebA dataset is used for a hair-color prediction task with facial images of celebrities, where the target labels are "blond" and "non-blond" hair colors. For experimental purposes, the dataset is divided into four distinct groups based on hair color and gender: $g_1 = \{$non-blond hair, female$\}$, $g_2 = \{$non-blond hair, male$\}$, $g_3 = \{$blond hair, female$\}$, and $g_4 = \{$blond hair, male$\}$. Gender serves as a spurious feature, introducing correlations between the hair color and gender of individuals. The training set consists of 162,770 images distributed as follows: 71,629 females with non-blond hair, 66,874 males with non-blond hair, 22,880 females with blond hair, and 1,387 males with blond hair. The validation set includes 19,867 images, with 8,535 females with non-blond hair, 8,276 males with non-blond hair, 2,874 females with blond hair, and 182 males with blond hair. The test set comprises 19,962 images, with 9,767 females with non-blond hair, 7,535 males with non-blond hair, 2,480 females with blond hair, and 180 males with blond hair. The minority group in this dataset is males with blond hair, which constitutes a small fraction of the data, highlighting the skewed distribution and the presence of spurious correlations.

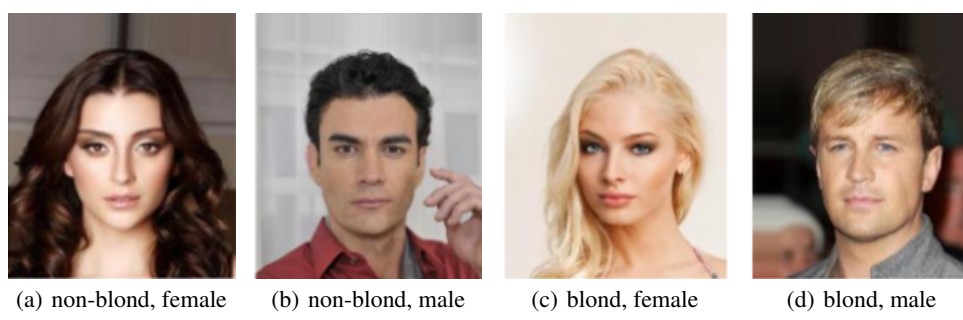

| (a) non-blond, female | (b) non-blond, male | (c) blond, female | (d) blond, male |

Figure 4: Example images from the CelebA dataset. The groups are $g_1 = \{$non-blond hair, female$\}$, $g_2 = \{$non-blond hair, male$\}$, $g_3 = \{$blond hair, female$\}$, and $g_4 = \{$blond hair, male$\}$.

### D.2 Modified Datasets

Building on the previously introduced datasets—CMNIST, Waterbirds, and CelebA—we constructed modified versions of these datasets by applying conditional distribution shifts to the minority groups, simulating real-world scenarios. Below, we detail the modifications for each dataset and illustrate these shifts with corresponding figures.

**Modified CMNIST**. In the CMNIST dataset, we created a modified version where the minority group's images (label 1, red) were rotated by 90 degrees in the test set, while they remained unrotated in the training set. This manipulation simulates conditional distribution shifts often encountered in real-world applications. Figure 5 provides an illustration of this shift, showing example images from the train and test sets.

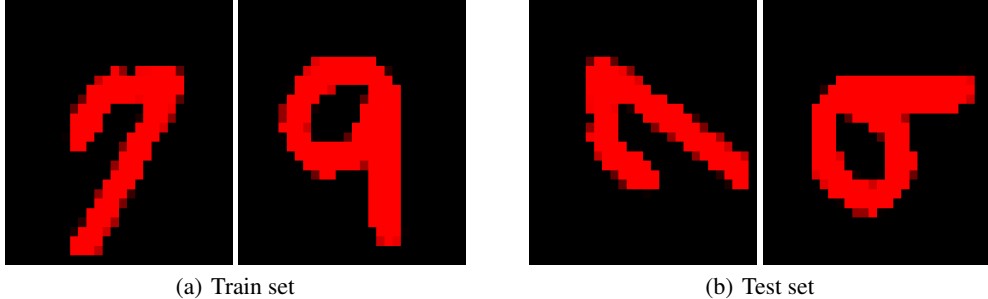

|(a) Train set|(b) Test set|

Figure 5: Example of conditional distribution shift in the CMNIST dataset, where the minority group (label 1, red) images are rotated by 90 degrees in the test set, while they are unrotated in the training set.

**Modified Waterbirds**. For the Waterbirds dataset, we constructed a modified version where the minority group (waterbird, land background) was designed to have a shift in species composition between the train and test sets. Specifically, the training set included only waterfowl species, such as Gadwall, Grebe, Mallard, Merganser, and Pacific Loon, while the test set contained exclusively seabird species, including Albatross, Auklet, Cormorant, Frigatebird, Fulmar, Gull, Jaeger, Kittiwake, Pelican, Puffin, Tern, and Guillemot. During the dataset construction process, we identified and corrected a mislabeling issue involving three species—Western Wood-Pewee, Eastern Towhee, and Western Meadowlark—which had been incorrectly labeled as waterbirds instead of landbirds [3]. Figure 6 illustrates this shift, highlighting the separation of species between the train and test sets.

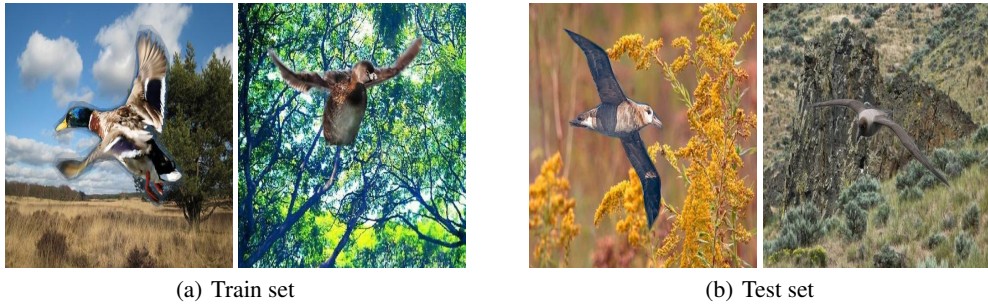

|(a) Train set|(b) Test set|

Figure 6: Example of conditional distribution shift in the Waterbirds dataset, where the minority group (waterbird on land background) consists of waterfowl in the training set and seabirds in the test set.

To further highlight the impact of this modification, Figure 7 compares the original distribution and the modified distribution shift scenarios. In the original dataset (Figure 7(a)), bird species in the minority group are relatively evenly distributed across train, validation, and test sets. However, in the modified version (Figure 7(b)), the training set contains only waterfowl, while the test set is composed entirely of seabirds, creating a distinct distribution shift.

**Modified CelebA**. In the CelebA dataset, we modified the minority group (blond hair, male) to have different attributes between the train and test sets. Specifically, the training set contained only images without glasses, while the test set contained only images with glasses. This modification reflects real-world distribution shifts where rare attributes in small minority groups may change across different distributions, impacting model performance. Figure 8 shows example images demonstrating this shift.

Figure 9 provides a detailed comparison of the original and modified distributions for the CelebA dataset. In the original distribution (Figure 9(a)), the minority group is predominantly represented by the "Without Eyeglasses" category across train, validation, and test sets, with relatively few examples in the "With Eyeglasses" category. In the modified version (Figure 9(b)), the training set consists

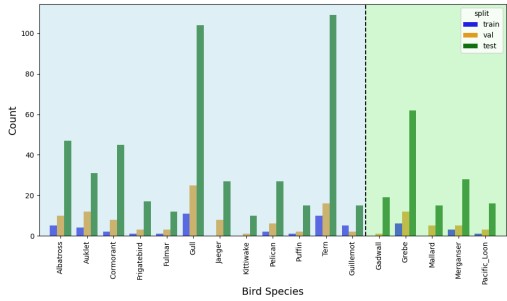

(a) Original distribution of the minority group.

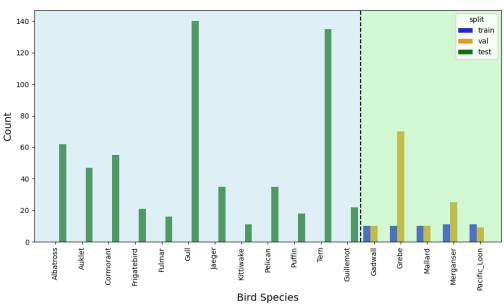

(b) Distribution shift of the minority group (only waterfowl in training, only seabirds in testing).

Figure 7: Comparing the original and shifted distributions of the minority group (waterbird, land background) in the Waterbirds dataset (left: seabirds, right: waterfowl, split by dashed line).

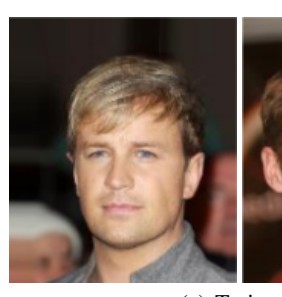 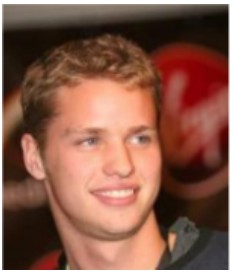

(a) Train set

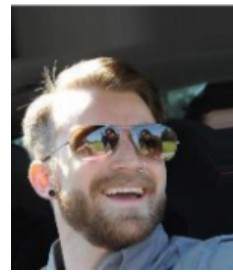 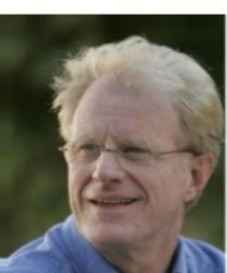

(b) Test set

Figure 8: Example of conditional distribution shift in the CelebA dataset, where the minority group (blond hair, male) included only images without glasses in the training set and images with glasses in the test set.

exclusively of "Without Eyeglasses" images, while the test set contains only "With Eyeglasses", creating a clear disjoint in key attributes between training and testing phases.

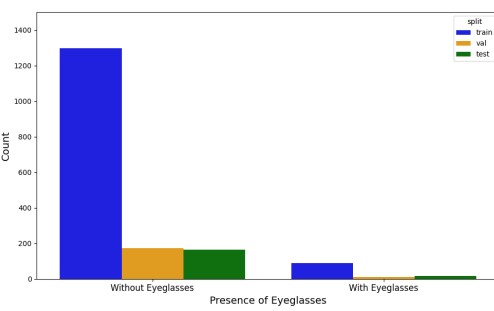

(a) Original distribution of the minority group.

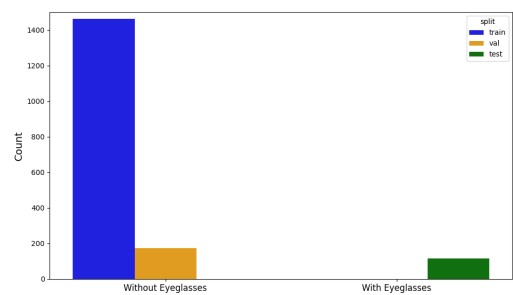

(b) Distribution shift of the minority group (only "Without Eyeglasses" in training, only "With Eyeglasses" in testing).

Figure 9: Comparing the original and shifted distributions of the minority group (blond hair, male) in the CelebA dataset (left: "Without Eyeglasses", right: "With Eyeglasses").

By introducing these conditional distribution shifts, our modified datasets simulate real-world challenges, particularly in scenarios where small minority groups are highly susceptible to such changes. These constructions not only reflect practical settings but also provide realistic benchmarks for evaluating the robustness and generalization capabilities of machine learning models under diverse and challenging conditions.

# E  Baseline Details

We compare our method against a range of representative baselines:

- **ERM**: ERM optimizes average accuracy on the training set without any robust objective or group-specific considerations.
- **Group DRO** [42]: A canonical approach for mitigating spurious correlations using known group labels. By partitioning data into predefined groups and minimizing the worst-case group loss, Group DRO aims to improve the worst-group accuracy relative to standard ERM.
- **JTT** [30]: A two-step method that first trains an ERM model to identify misclassified samples (viewed as proxies for minority groups), then upsamples these samples and retrains a classifier.
- **CnC** [56]: Identifies samples that share the same true class but differ in spurious attributes by analyzing ERM outputs, then trains a robust model with a contrastive learning objective. This does not require explicit group labels.
- **SAA** [35]: Infers latent groups via a loss-based criterion, then applies Group DRO to improve robustness. This method partially automates the discovery of group boundaries without needing full group labels.
- **LISA** [53]: Mitigates spurious correlations by using Mixup strategies. Depending on the dataset, LISA employs different Mixup variants (e.g., classic Mixup, CutMix, Manifold Mix) to interpolate images within the same label or same spurious attribute, thereby reducing reliance on superficial cues.
- **DFR** [24]: Balances the dataset by subsampling to match the minority group size (the "Subsample" strategy), then retrains an ERM model on this balanced data. This simple yet effective approach can substantially improve worst-group performance. For a fair comparison, following [12], we evaluate DFR using only the training dataset for both training and fine-tuning, ensuring consistency across methods, which is denoted as DFR$^{\text{tr}}$.
- **PDE** [12]: Progressively expands the training dataset during the training process, starting with a balanced subset to prevent the model from learning spurious correlations. This approach aims to enhance robustness across all groups, including underrepresented ones.
- **GIC** [19]: Uses a two-step pipeline where group membership is partially inferred, then a robust optimization (e.g., Group DRO) is applied. Similar to LISA, it can incorporate tailored Mixup strategies depending on the dataset's characteristics.

# F  Implementation Details

For experiments involving our newly constructed datasets, we reimplemented both our proposed method and the relevant baselines. When certain baselines lacked reported results for a given dataset, we used the performance from [56] and [19] if available; otherwise, we performed our own reimplementations under consistent settings. In particular, for the original CMNIST dataset, we reimplemented experiments for DFR and PDE, since their original papers did not include CMNIST results. In all other cases, we referenced performance metrics from each baseline's primary source. All experiments were conducted on an NVIDIA GeForce RTX 3090 GPU.

Across all datasets, we employed the torchvision implementation of ResNet-50 pretrained on ImageNet, training with SGD at a momentum of 0.9 and a batch size of 128, following [42]. Our approach also introduces a perturbation parameter $\epsilon$ to control within-group uncertainty. Specifically, we define $\epsilon_g = \epsilon/\sqrt{n_g}$, where $n_g$ represents the size of group $g$ in the training data. To determine $\epsilon$, we performed a grid search over the set $\{12/255, 24/255, 36/255, 48/255, 60/255, 72/255, 84/255, 96/255\}$, scaling each value by $\sqrt{n_{\min}}$. Here, $n_{\min} = \min_g n_g$ denotes the smallest group size in the training set. Additionally, we tuned the generalization adjustment parameter $C$ over $\{0, 1, 2, 3\}$, as described in Section 3.3 of [42]. This setup was applied consistently across every dataset.

For CMNIST, we conducted a grid search over learning rates $\{10^{-4}, 10^{-3}, 10^{-2}\}$ and $\ell_2$ penalties $\{10^{-1}, 10^{-2}, 10^{-4}\}$ for 50 epochs. Due to instability in training with the selected parameter combinations in the original Group DRO implementation, we applied a ReduceLROnPlateau scheduler starting at a learning rate of 0.01, using it consistently for both our method and Group DRO to ensure fairness. For Waterbirds, the learning rate was tuned over $\{10^{-3}, 10^{-4}, 10^{-5}\}$ and the $\ell_2$ penalty

over $\{10^{-4}, 10^{-1}, 1\}$, with training conducted for 300 epochs. For CelebA, the learning rate was tuned over $\{10^{-4}, 10^{-5}\}$ and the $\ell_2$ penalty over $\{10^{-4}, 10^{-2}, 1\}$ for 30 epochs. We referred to prior works including [53] and [17] to guide these hyperparameter search ranges.

# G Experimental Result Details

## G.1 Visualizing t-SNE Ordering for Minority Groups

To highlight the utility of t-SNE ordering in simulating realistic distribution shifts (as discussed in Section 4.3), we present visual examples of waterbirds from the top 20% and bottom 20% quantiles of the t-SNE projections. This approach effectively partitions samples based on semantically meaningful intra-group differences, enabling validation splits that closely mimic real-world distribution shifts.

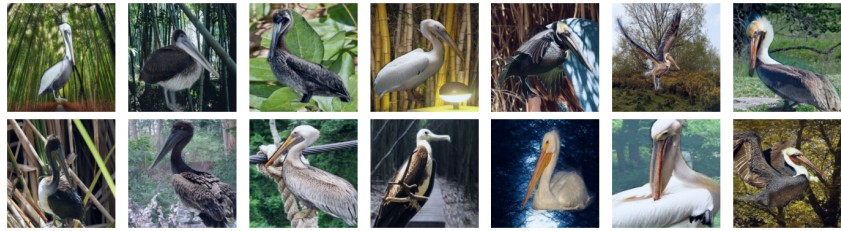

(a) Example images from the top 20% of t-SNE ordering

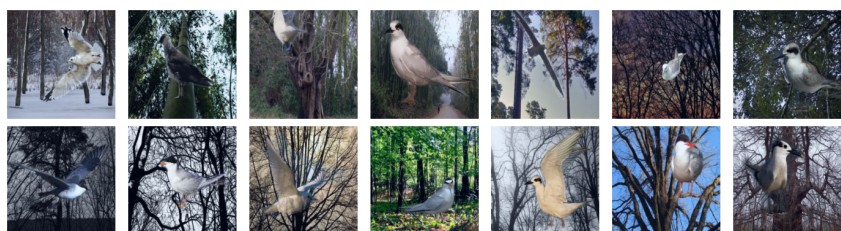

(b) Example images from the bottom 20% of t-SNE ordering

Figure 10: t-SNE-based ordering reveals subtle distinctions within the minority group. (a) The top quantile features waterbirds with longer beaks, while (b) the bottom quantile features those with shorter beaks.

As shown in Figure 10, the t-SNE ordering captures nuanced intra-group differences within this minority group. In the top 20% quantile (Figure 10(a)), waterbirds with longer beaks dominate; in the bottom 20% quantile (Figure 10(b)), shorter-beaked waterbirds are more prevalent. This contrast illustrates how a t-SNE-driven partition can create validation splits that mimic real-world distribution shifts. This method not only emphasizes variations within groups but also systematically evaluates the model's robustness under challenging real-world conditions.

## G.2 Impact of $\epsilon$ on Robustness

The perturbation parameter $\epsilon$ plays a critical role in improving robustness under minority group shifts. Figures 11(a) and 11(b) show how increasing $\epsilon$ affects worst-group accuracy for the Waterbirds and CelebA datasets, respectively. Notably, both datasets achieve significant gains in worst-group accuracy when $\epsilon$ is set above zero, indicating enhanced resilience to distributional shifts.

As illustrated in Figure 11(a), larger $\epsilon$ values consistently improve worst-group accuracy on Waterbirds, enabling the model to better manage intra-group variations and subpopulation shifts. A similar trend appears in Figure 11(b) for CelebA, further validating the robustness gained by appropriately increasing $\epsilon$.

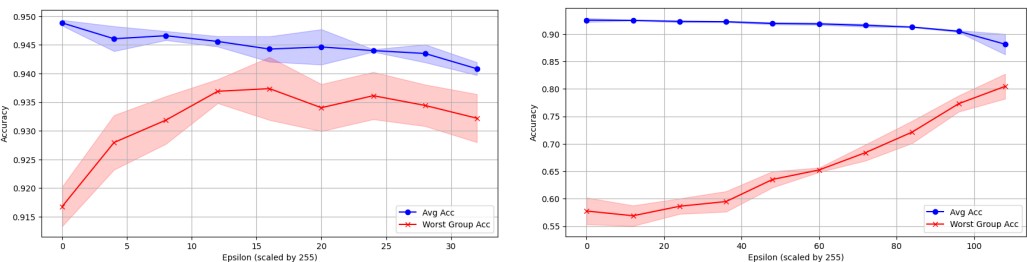

(a) Waterbirds dataset under a minority group shift.  (b) CelebA dataset under a minority group shift.

Figure 11: Impact of $\epsilon$ on robustness. The x-axis represents $\epsilon$ values scaled by 255, and the y-axis indicates accuracy. Each point is the mean of 3 runs (solid lines), and the shaded regions show the standard deviation. For this analysis, the learning rate and $\ell_2$ penalty were fixed to isolate the effect of $\epsilon$.

These findings underscore the importance of incorporating conditional distribution uncertainty into the training framework. By effectively capturing within-group variability, our approach significantly enhances worst-group performance, making it well-suited for handling realistic distributional shifts.

### G.3 Grad-CAM Results and Analysis

To gain further insight into where each model focuses its attention under minority-group shifts, we visualize Grad-CAM [43] heatmaps on misclassified examples (by Group DRO) that our method classifies correctly. Figure 12 shows examples on the Waterbirds dataset, while Figure 13 presents examples from CelebA.

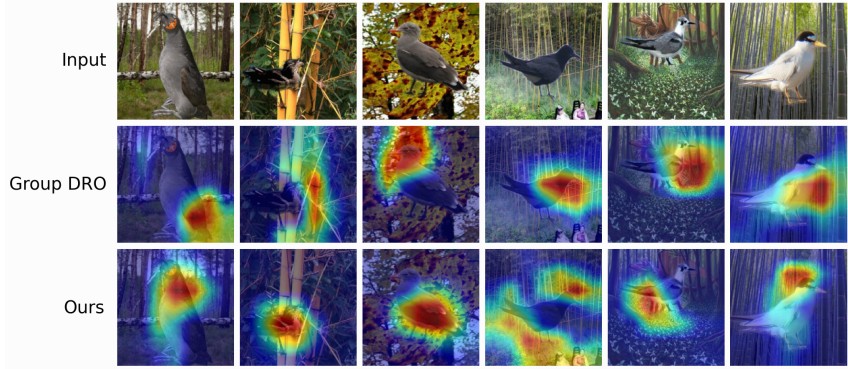

Figure 12: Grad-CAM visualizations for Waterbirds test images from a minority-group shift scenario. Each column shows an input image (top row), Grad-CAM for Group DRO (middle row), and Grad-CAM for our method (bottom row).

**Waterbirds.** In Figure 12, the minority-group shift involves species changes not observed in the training set. While Group DRO often localizes on a narrow region of the bird—sometimes near the torso or background—our method exhibits a more distributed attention, covering details like the wings, beak, or feet. This broader localization helps the model rely on features invariant to previously unseen waterbird species, enabling robust classification despite changes in the specific types of waterbirds encountered.

**CelebA.** Figure 13 shows examples from the minority group (blond-hair, male) in which the test images include glasses—an attribute absent from the training set. In these cases, Group DRO erroneously directs attention toward the facial or eyewear regions rather than focusing on hair color. By contrast, our method more reliably highlights the hair region, aligning with the intended classification objective and enabling correct predictions even under previously unseen attributes.

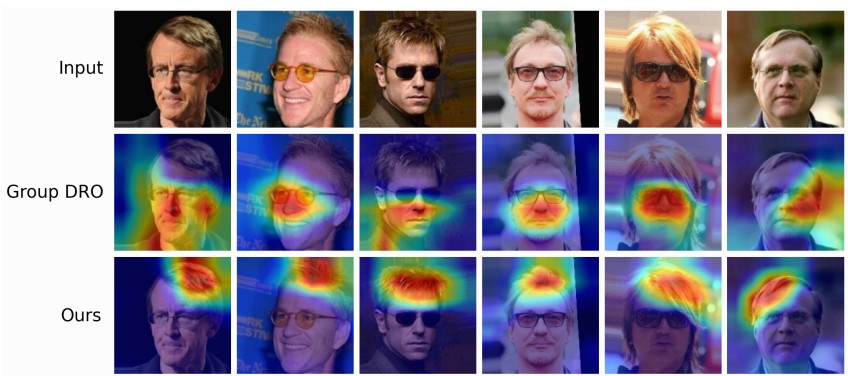

Figure 13: Grad-CAM visualizations for CelebA test images from the minority-group shift scenario. Each column shows the input image (top row), Grad-CAM for Group DRO (middle row), and Grad-CAM for our method (bottom row).

Overall, these visualizations confirm that, under challenging distribution shifts, our hierarchical DRO framework is less prone to confounding features and more successful in focusing on the task-relevant regions. This broader and more contextually aligned attention helps maintain strong performance even when encountering unseen or spurious attributes.

