# OpenReview forum: "Mitigating Spurious Correlation via Distributionally Robust Learning with Hierarchical Ambiguity Sets"
_NeurIPS.cc/2025/Conference — Submitted to NeurIPS 2025_

### Official Review · Reviewer_y2Kb · 2025-06-22

**Clarity:** 3
**Significance:** 3
**Originality:** 3
**Rating:** 5
**Confidence:** 3

**Summary:**

The authors argue that current approaches to tackling spurious correlations and distribution shifts assume "clean" groups and do not account for intra-group variations. To tackle this problem, they propose a hierarchical version of GroupDRO, which models both the mixing proportions between groups via $\beta$ weights belonging to the $\Delta_{m-1}$ simplex, where $m$ is the number of groups and the intra-group variability via a second constraint on the ambiguity set, modeled via Wasserstein distances and a cost function based on the distance between representations in the latent space.

To make the problem tractable, they optimize a surrogate objective that corresponds to an upper bound on the actual loss, arguing that the hierarchical ambiguity set can be modeled (or over-approximated) using adversarial perturbations in the latent space. In practice, for a single sample $x_i$ they:

1. Update the adversarial perturbation $z_i$ via a single gradient ascent step, bounding it to a sphere defined by $\epsilon_g$ via projections as needed.
2. Update the weights $\beta$
3. Update the model parameters via SGD.

The hyper-parameter $\epsilon_g$ is picked by:

1. Encoding the group training samples.
2. Projecting the embeddings to one dimension via t-SNE.
3. Splitting the samples into 5 quantiles and then training models using either the bottom 4 or top 4 quantiles and the remaining quantile as a validation set. The authors argue that this simulates realistic distribution shifts for minority groups.
4. They then average the two runs and pick the $\epsilon_g$ parameter that maximizes the minority-group accuracy on average.

Their method is then evaluated on standard benchmarks with group annotations, namely CMNIST, Waterbirds, and CelebA, and on newly introduced variations of them that model intra-group shifts:

1. Shifted CMNIST, where all images in the minority group (label 1, red color) are rotated by 90 degrees at test time.
2. Shifted Waterbirds, where the training set minority group (waterbird, land background) only contains waterfowls, and the test set only seabirds.
3. Shifted CelebA, where the minority group (blonde, male) only has images without glasses during training and with glasses at test time.

On shifted benchmarks, their method consistently achieves the best performance, improving by up to ~15% over GroupDRO and more than other baselines that utilize labels.

On standard benchmarks, they achieve performances that are either slightly better than or equal to GroupDRO, both for the group with the worst accuracy and for the average accuracy. Similarly, they outperform every other baseline that either uses group annotations or generates its own, with the exception of PDE on CelebA. Still, they largely surpass it in a shifted setup.

**Questions:**

1. It would be interesting to know the training runtime of the newly proposed method compared to GroupDRO and ERM.

**Ethical Concerns:**

["NO or VERY MINOR ethics concerns only"]

**Final Justification:**

The authors adequately addressed two out of my three concerns. I am thus happy to increase the score to 5.

**Limitations:**

The authors properly addressed the limitations of their work.

**Paper Formatting Concerns:**

N/A.

**Quality:**

2

**Strengths And Weaknesses:**

### Strengths

1. The paper is well-written and easy to follow.
2. The paper tackles an important problem, namely the performance of machine learning models in the presence of spurious correlations and distribution shifts. Furthermore, the paper goes further than traditional group annotations, by considering the problem of distribution shifts within groups. These shifts are modeled by altering existing and well-known benchmark datasets.
3. The method is reasonably simple and intuitive, and leads to performance improvements on both traditional benchmarks with "clean" groups and on the newly introduced benchmarks. On the latter, the proposed method improves by up to 15% over GroupDRO, and more over other baselines that make use of group annotations.

### Weaknesses

1. While the authors clearly explain their experimental setup, models, and hyper-parameter ranges used in their experiment, the exact configuration(s) for each dataset are missing. Reporting those details would facilitate reproducibility and make it easier and faster. Similarly, the authors mention that they use Torchvision's ResNet-50 model, but do they mean the version with IMAGENET1K_V1 or IMAGENET1K_V2 weights? Is this the exact same model used by other methods whose performance was reported from other papers, e.g. [59] and [19]? I am not super confident on it being the case, as e.g. [59] is from ICML2021 (Summer 2021), while the IMAGENET1K_V2 weights (the current default) were published in November 2021 according to the blog post linked on the model page.
2. The evaluation is limited to datasets with a small number of label/group combinations. It would be interesting to see how well this method scales as the number of label/group combinations grows, e.g. on iWildCam2020-WILDS.

I am willing to increase my score in case the doubts regarding the ResNet-50 backbone can be clarified.

---

> ### Author Rebuttal · Authors · 2025-07-31
>
> We sincerely appreciate the reviewer’s constructive feedback and careful reading of our paper.
>
> ---
>
> **W1**
>
> > While the authors clearly explain their experimental setup, models, and hyper-parameter ranges used in their experiment, the exact configuration(s) for each dataset are missing. Reporting those details would facilitate reproducibility and make it easier and faster. Similarly, the authors mention that they use Torchvision's ResNet-50 model, but do they mean the version with `IMAGENET1K_V1` or `IMAGENET1K_V2` weights? Is this the exact same model used by other methods whose performance was reported from other papers, e.g. [59] and [19]? I am not super confident on it being the case, as e.g. [59] is from ICML2021 (Summer 2021), while the `IMAGENET1K_V2` weights (the current default) were published in November 2021 according to the blog post linked on the model page.
>
> We thank the reviewer for this helpful comment. We agree that reporting dataset-specific configurations more explicitly would improve reproducibility. While Appendix D.2 already describes the construction details of the shifted datasets, we acknowledge that releasing the code used to generate these splits would further facilitate replication and experimentation. We will therefore include the dataset construction scripts in our revised version.
>
> Regarding the ResNet-50 backbone, we verified the exact version used in our implementation. Specifically, we used:
>
> `
> torchvision.models.resnet50(pretrained=True)
> `
>
> During execution, PyTorch emits the following warning:
>
> ```python
> UserWarning: Arguments other than a weight enum or None for 'weights' are deprecated since 0.13 and may be removed in the future. The current behavior is equivalent to passing weights=ResNet50_Weights.IMAGENET1K_V1. You can also use weights=ResNet50_Weights.DEFAULT to get the most up-to-date weights.
> ```
> This confirms that our model uses the V1 weights, which aligns with the setup used in prior works.
>
> We appreciate the reviewer’s attention to this versioning detail. Although we did not explicitly consider the difference between V1 and V2 weights during implementation, we recognize its relevance and will be more mindful of such changes in future work.
>
>
>
> ---
>
> **W2**
> >The evaluation is limited to datasets with a small number of label/group combinations. It would be interesting to see how well this method scales as the number of label/group combinations grows, e.g. on iWildCam2020-WILDS.
>
> Thank you for the insightful suggestion. We agree that evaluating how our method scales with an increasing number of label/group combinations is an important direction.
>
> In this work, we focused on datasets that are widely used in the spurious correlation literature, to ensure comparability with prior work and to highlight challenges specific to intra-group distribution shifts. That said, we fully agree that validating performance in more complex settings—such as those with a larger number of groups—is crucial for assessing broader applicability.
>
> Although we have not included experiments on datasets like iWildCam2020-WILDS, we consider this a meaningful next step and plan to explore it in future work.
>
> ---
>
> **Q1**
> >It would be interesting to know the training runtime of the newly proposed method compared to GroupDRO and ERM.
>
> Thank you for the question. Although our method introduces additional modeling components—namely, the definition of group-wise ambiguity sets in the latent space—it remains lightweight in practice.
>
> We perform only a single-step gradient ascent to compute the inner maximization and apply perturbations in the latent representation. This avoids full backpropagation through the entire network and instead operates only up to the final classification layer. In addition, since the latent space is much lower in dimensionality than the raw input (e.g., $2048$ vs. $3 \times 224 \times 224$), the associated cost remains minimal. As a result, the method results in only about $1.3\times$ the training time of standard GroupDRO.
>
> We will include a discussion of runtime efficiency in the revised version, as this point was brought up by multiple reviewers. Clarifying these details will help underscore the method’s efficiency and reinforce its practical relevance.
>
> ---
>
> We thank the reviewer again for the constructive and detailed feedback, as well as for pointing out practical aspects—such as model configuration and reproducibility—that can further strengthen our work. We hope these clarifications address the reviewer’s concerns.

---

> > ### Comment · Reviewer_y2Kb · 2025-08-02
> >
> > Thank you for your rebuttal! I have increased my score as promised since you have addressed two out of three concerns.

---

### Official Review · Reviewer_drEy · 2025-06-27

**Clarity:** 3
**Significance:** 2
**Originality:** 2
**Rating:** 3
**Confidence:** 3

**Summary:**

This paper proposes a method to address spurious correlations, building on top of ideas from Group DRO by adding a hierarchical ambiguity set that allows for changing the intra-group mixture in case each group's conditional distributional is not captured. This results in a min-max objective that is is approximated by an inner gradient-ascent step on a latent feature and a mirror-descent update on group weights. The authors test on various vision benchmarks with synthetic spurious correlations.

**Questions:**

See weaknesses.

**Ethical Concerns:**

["NO or VERY MINOR ethics concerns only"]

**Final Justification:**

I am still not convinced of the method's applicability to naturally occurring real-world settings. Since the main contributions of this paper are empirical in nature, I think the paper could benefit from showing this better in a future revision of the paper.

**Limitations:**

yes

**Quality:**

2

**Strengths And Weaknesses:**

Strengths:
- The method leads to better performance when there are severe intra-group shifts and minority subgroup alteration.
- The framework is a natural extension of prior DRO variants.

Weaknesses:
- The only significant performance gain occurs with the author-devised shifted splits. It would be good to show that this occurs in more complex benchmarks or real-world datasets. There are no experiments on more difficult benchmarks, and the method does not improve performance on the standard benchmarks. Can you demonstrate benefits on more realistic benchmarks and improve the thoroughness of the evaluation?
- The method adds quite a bit of extra computational overhead along with extra hyperparameters, e.g epsilon is chosen in a bit of an adhoc manner. It seems like a lot of extra complexity that might not be that convincing to use in practice and might not generalize to more realistic situations. Can you quantify the time and memory increase relative to Group DRO?

---

> ### Author Rebuttal · Authors · 2025-07-31
>
> We thank the reviewer for the valuable feedback and the opportunity to clarify.
>
> ---
>
> **W1**
> >The only significant performance gain occurs with the author-devised shifted splits. It would be good to show that this occurs in more complex benchmarks or real-world datasets. There are no experiments on more difficult benchmarks, and the method does not improve performance on the standard benchmarks. Can you demonstrate benefits on more realistic benchmarks and improve the thoroughness of the evaluation?
>
> We would like to note that our experiments build upon standard and widely adopted benchmarks in the spurious correlation literature, including CelebA, Waterbirds, and CMNIST. These datasets have been consistently used in prior works due to their clear group structure and tractable spurious features, and are commonly considered representative of real-world robustness challenges in structured group settings [1, 2, 3]. In this regard, our work represents an effort to extend the evaluation landscape of standard benchmarks by introducing more realistic and complex intra-group shift scenarios—in fact, moving beyond the conventional settings used in many recent works.
>
> To design intra-group shifts, we intentionally selected perturbations based on existing dataset attributes (e.g., "glasses" in CelebA), which we believe is a principled approach to ensure objectivity and reproducibility. In fact, we initially explored using other attributes such as “wavy hair” or “beard” but found them too noisy or inconsistent for controlled intra-group shift evaluation.
>
> Importantly, we view the construction of these modified splits as a contribution in itself. Our proposed splits reveal an important limitation of prior approaches—namely, their sensitivity to intra-group distribution shifts, even when induced by simple changes to the train/test partition. This setup enables a clear and controlled demonstration of both the failure modes of baseline methods and the effectiveness of our proposed solution. Notably, this line of work aligns with recent efforts to stress-test existing benchmarks through controlled modifications. For example, [4] introduced the MultiCelebA dataset—an extension of CelebA with multiple spurious attributes—to study robustness under more complex spurious correlation settings. We view our effort as part of a broader research direction aimed at building models that generalize beyond fixed data partitions and truly capture robustness in the face of structural variation.
>
> That said, we fully agree that it is important to evaluate robustness beyond the attribute-based perturbations used in our current benchmarks. To address this, we conducted a synthetic experiment where intra-group shifts are defined by latent shifts in the core features of the data-generating distribution. This toy example serves to test generalization in settings without semantic attributes and allows for a clearer understanding of our method’s behavior under structured latent shifts.
>
> Specifically, we construct a 3D feature space with four group-conditional Gaussian distributions. Labels are determined by the core features $x_1 + x_2 > 6$, while the third feature $x_3$ acts as a spurious attribute:
>
> - $X_{\text{group1}} \sim \mathcal{N}([6, 6, 10]^T, \mathbf{I})$, label 0 (4000 samples)
> - $X_{\text{group2}} \sim \mathcal{N}([-2, -2, 10]^T, \mathbf{I})$, label 1 (100 samples)
> - $X_{\text{group3}} \sim \mathcal{N}([8, 8, 0]^T, \mathbf{I})$, label 0 (100 samples)
> - $X_{\text{group4}} \sim \mathcal{N}([0, 0, 0]^T, \mathbf{I})$, label 1 (4000 samples)
>
>
> At test time, we shift the core dimensions of the minority groups while keeping the spurious dimension fixed:
> Group 2 shifts from $[-2, -2]^T$ to $[1.5, 1.5]^T$, and Group 3 from $[8, 8]^T$ to $[4.5, 4.5]^T$.
>
> | **Method**   | **Group 1** | **Group 2** | **Group 3** | **Group 4** |
> |--------------|-------------|-------------|-------------|-------------|
> | Group DRO    | 100.0%      | 87.6%       | 81.8%       | 100.0%      |
> | Ours         | 100.0%      | 99.1%       | 98.6%       | 100.0%      |
>
>
> These results indicate that our method improves robustness even under structural distribution shifts without relying on pre-defined semantic attributes. We will include this additional discussion and experimental result in the revised version to better emphasize the generalizability and practical value of our approach.
>
> ---
>
> **W2**
> >The method adds quite a bit of extra computational overhead along with extra hyperparameters, e.g epsilon is chosen in a bit of an adhoc manner. It seems like a lot of extra complexity that might not be that convincing to use in practice and might not generalize to more realistic situations. Can you quantify the time and memory increase relative to Group DRO?
>
> We appreciate the reviewer’s concern regarding the added complexity of our method. We would like to address both the computational overhead and the role of hyperparameter tuning in detail.
>
> First, while our method introduces additional modeling flexibility—specifically, by defining ambiguity sets using Wasserstein distances computed in latent space—it remains computationally efficient in practice. As shown in Algorithm 1, we use a single-step gradient ascent to compute the inner maximization and apply perturbations in the latent space. This only requires backpropagation up to the final layer, avoiding the need for full backpropagation through the encoder. Moreover, the latent space (e.g., $2048$ dimensions) is significantly lower-dimensional than the input space (e.g., $3 \times 224 \times 224$), which further reduces the cost. As a result, our method incurs only about $1.3×$ the training time of standard GroupDRO.
>
> Second, regarding the $\epsilon$ parameter: we acknowledge that it introduces an extra tuning step, but this is a standard and widely accepted aspect of robust learning frameworks—including distributionally robust optimization in domain adaptation and adversarial training. In fact, it is not uncommon in those settings to either select $\epsilon$ arbitrarily or simply report performance over a range of values (as we show in Appendix G.2). In contrast, we aimed to offer a simple and effective guideline for selecting $\epsilon$ in a realistic setting, without relying on target shift information.
>
> In line with Reviewer 7mur’s question (Q1), we maintained the original train/validation split and intentionally avoided applying test-time perturbations to the validation set. This design choice was made to ensure the fairness of evaluation, rather than tuning to a known or artificially aligned distribution.
>
> Finally, we agree that clarifying these practical considerations would strengthen the paper. To that end, we will revise the manuscript to reflect the above discussion on training efficiency and hyperparameter tuning, with the aim of improving reproducibility and real-world applicability.
>
> ---
>
> We thank the reviewer once again for the feedback. We hope our responses have clarified the motivation behind our design choices and the practical relevance of our method. We also hope this exchange has helped better convey the novelty and contribution of our work.
>
> ---
>
> **Reference**
>
> [1] Yao et al., Improving out-of-distribution robustness via selective augmentation, ICML 2022
>
> [2] Kirichenko et al., Last layer re-training is sufficient for robustness to spurious correlations, ICLR 2023
>
> [3] Deng et al., Robust learning with progressive data expansion against spurious correlation, NeurIPS 2024
>
> [4] Kim et al., Improving Robustness to Multiple Spurious Correlations by Multi-Objective Optimization, ICML 2024

---

> > ### Comment · Reviewer_drEy · 2025-08-05
> > **Rebuttal response**
> >
> > Thanks to the authors for the rebuttal. I am still not convinced of the method's applicability to naturally occurring real-world settings. Since the main contributions of this paper are empirical in nature, I think the paper could benefit from showing this better in a future revision of the paper, so I will keep my score.

---

> ### Author Response · Authors · 2025-08-06
>
> Thank you for your follow-up and for engaging with our rebuttal.
>
> We understand and appreciate your continued concern regarding real-world applicability. In response, we would like to reiterate that our work addresses a practically important but underexplored phenomenon: when a minority group is too small to reliably reflect its true underlying distribution, widely used methods such as GroupDRO can fail—even when group annotations are available. This setting is statistically well-grounded, as small sample sizes naturally lead to higher variance and less reliable estimates of the underlying distribution.
>
> We would also like to emphasize that identifying new failure modes within the spurious correlation framework is an actively growing research direction. For instance, recent studies have investigated imperfect group partitions [1], multiple spurious features [2, 3], and low spurious signal strength [4]. Like ours, these works highlight new challenges by modifying standard benchmark datasets rather than constructing entirely new ones—a practice widely accepted in the literature for ensuring comparability and reproducibility.
>
> In fact, we believe this represents a fair and realistic way to evaluate new methods within the spurious correlation setting, precisely because it builds directly on widely adopted datasets. This evaluation strategy is supported by the design choices made in these prior studies and aligns with the broader goals of robustness research.
>
> We would also like to note that some of these works (e.g., [3, 4]) do not apply their modified settings across all datasets (e.g., no modified version for Waterbirds), whereas our evaluation remains more consistent and uniform across benchmarks. We believe this consistency further strengthens the empirical grounding of our approach.
>
> We fully agree that demonstrating real-world relevance is important, and we hope that our framing—together with evaluation choices aligned with prevailing community standards—helps clarify the broader value of our contribution.
>
> We hope this explanation helps address your concerns, and we sincerely appreciate your thoughtful feedback.
>
> ---
>
> **Reference**
>
> [1] Zhou et al., Examining and Combating Spurious Features under Distribution Shift, ICML 2021.
>
> [2] Paranjape et al., AGRO: Adversarial Discovery of Error-prone groups for Robust Optimization, ICLR 2023.
>
> [3] Kim et al., Improving Robustness to Multiple Spurious Correlations by Multi-Objective Optimization, ICML 2024.
>
> [4] Mulchandani and Kim, Severing Spurious Correlations with Data Pruning, ICLR 2025.

---

### Official Review · Reviewer_PVZF · 2025-06-30

**Clarity:** 3
**Significance:** 2
**Originality:** 2
**Rating:** 5
**Confidence:** 3

**Summary:**

This paper proposes a hierarchical extension of Group DRO that addresses both inter-group and intra-group uncertainties. It also introduces new benchmark settings that simulate realistic minority-group distribution shifts. Experiments show that the proposed method exhibits robustness under conditions where existing robust learning methods fail, while also achieving good performance on standard benchmarks.

**Questions:**

- Can the proposed method be applied to other Group DRO variants?
- What is the computational cost of the method?
- The variance of the baseline methods on the shifted datasets is somewhat high (e.g., ±8.2 points on LISA). What causes this variability, and could it undermine the reported comparisons? Why does the proposed method not experience this high variance?

**Ethical Concerns:**

["NO or VERY MINOR ethics concerns only"]

**Final Justification:**

The author addressed my concerns with a detailed response. I also read the review from other reviewers and better understood the paper. I will keep my positive score for the paper.

**Limitations:**

yes

**Quality:**

3

**Strengths And Weaknesses:**

**Strengths**

- The paper examines an important yet often overlooked question of both inter-group and intra-group distribution shifts. Current methods mostly focus on the inter-group distribution shift. The paper extends Group DRO to a hierarchical ambiguity set that captures both inter-group and intra-group uncertainties.
- The paper is well-written and easy to follow, with theoretical support clearly stated, datasets well explained, and experiments well illustrated.
- Extensive experiments are conducted to validate the method.
- The paper also establishes challenging evaluation scenarios to introduce minority-group distribution shifts that prior work has overlooked, with careful designs.

**Weaknesses**

- While the modified datasets are valuable, the intra-group shifts are somewhat specific and limited (e.g., rotation for CMNIST, species change for Waterbirds, glasses for CelebA). It would be beneficial to discuss the generalizability of the proposed method to other types of intra-group shifts that might be more complex, or to settings where such clear “attributes” for shifting are not readily available.

---

> ### Author Rebuttal · Authors · 2025-07-31
>
> Thank you for the thoughtful comments and helpful suggestions. We address each point below.
>
> ---
>
> **W1**
> >While the modified datasets are valuable, the intra-group shifts are somewhat specific and limited (e.g., rotation for CMNIST, species change for Waterbirds, glasses for CelebA). It would be beneficial to discuss the generalizability of the proposed method to other types of intra-group shifts that might be more complex, or to settings where such clear “attributes” for shifting are not readily available.
>
> We thank the reviewer for this thoughtful comment. In constructing our shifted datasets, we intentionally selected perturbations (e.g., glasses in CelebA) that rely on existing attributes in the datasets to ensure objectivity, reproducibility, and clarity in evaluating intra-group distribution shifts. For example, in CelebA, we initially considered using other attributes such as “wavy hair” or “beard”, but found them to be noisy or inconsistent across samples, which made controlled intra-group shift construction difficult.
>
> To further demonstrate the generalizability of our method beyond attribute-driven shifts, we conducted a controlled synthetic experiment using group-conditional Gaussians with intra-group variation in the core feature dimensions. This setup simulates distribution shifts at the level of data-generating processes, without relying on predefined semantic attributes. As shown below, our method achieves substantial gains in robustness under such latent intra-group shifts.
>
> Specifically, we construct a 3D feature space with four group-conditional Gaussian distributions. Labels are determined by the core features $x_1 + x_2 > 6$, while the third feature $x_3$ acts as a spurious attribute:
>
> - $X_{\text{group1}} \sim \mathcal{N}([6, 6, 10]^T, \mathbf{I})$, label 0 (4000 samples)
> - $X_{\text{group2}} \sim \mathcal{N}([-2, -2, 10]^T, \mathbf{I})$, label 1 (100 samples)
> - $X_{\text{group3}} \sim \mathcal{N}([8, 8, 0]^T, \mathbf{I})$, label 0 (100 samples)
> - $X_{\text{group4}} \sim \mathcal{N}([0, 0, 0]^T, \mathbf{I})$, label 1 (4000 samples)
>
> At test time, we shift the core dimensions of the minority groups while keeping the spurious dimension fixed:
> Group 2 shifts from $[-2, -2]^T$ to $[1.5, 1.5]^T$, and Group 3 from $[8, 8]^T$ to $[4.5, 4.5]^T$.
>
> | **Method**   | **Group 1** | **Group 2** | **Group 3** | **Group 4** |
> |--------------|-------------|-------------|-------------|-------------|
> | Group DRO    | 100.0%      | 87.6%       | 81.8%       | 100.0%      |
> | Ours         | 100.0%      | 99.1%       | 98.6%       | 100.0%      |
>
> These results indicate that our method remains effective even under distribution shifts arising from underlying structural variations within groups, supporting its broader applicability. We will incorporate this discussion into the revised version to more clearly highlight the generalizability of our method.
>
> ---
>
> **Q1**
> >Can the proposed method be applied to other Group DRO variants?
>
> Yes, our method can be applied to other Group DRO variants. In fact, Group DRO continues to serve as the foundation for a wide range of recent works. For example, several recent studies [1, 2, 3] build on the original Group DRO formulation.
>
> Our Method can be seamlessly integrated into such pipelines as a drop-in replacement for the Group DRO step. This flexibility allows our method to inherit the strengths of these advanced variants while adding robustness to intra-group distribution shifts.
>
> ---
>
> **Q2**
> >What is the computational cost of the proposed method?
>
> Thank you for the question. While our method introduces additional modeling flexibility---specifically by defining ambiguity sets using Wasserstein distances in latent space---it remains computationally efficient in practice.
>
> We use a single-step gradient ascent to compute the inner maximization and apply perturbations in the latent space. This requires only backpropagation up to the final layer of the network, avoiding the need for full backpropagation through the encoder. Moreover, since the latent space (e.g., $2048$ dimensions) is much lower-dimensional than the input space (e.g., $3 \times 224 \times 224$), the additional cost remains modest.
>
> As a result, our method incurs only about $1.3\times$ the training time of standard GroupDRO, while offering significantly improved robustness to intra-group shifts.
>
> As this point was noted by multiple reviewers, we will incorporate it into the revised version to clarify the method’s efficiency and practical relevance.
>
> ---
>
> **Q3**
> >The variance of the baseline methods on the shifted datasets is somewhat high (e.g., ±8.2 points on LISA). What causes this variability, and could it undermine the reported comparisons? Why does the proposed method not experience this high variance?
>
> Thank you for the question. We implemented all baseline methods using the suggested settings provided in their respective papers. The observed variance appears to reflect the sensitivity of these models when exposed to underrepresented subpopulations or severe distributional shifts. In such settings, small changes in initialization (e.g., due to different random seeds) can lead to disproportionately large variations in performance.
>
> In contrast, our method explicitly accounts for uncertainty within groups, which helps mitigate such variance. We believe this difference serves as indirect evidence of the robustness benefits of our approach, particularly in challenging regimes.
>
> ---
>
> We greatly appreciate your insightful comments and suggestions, which have contributed meaningfully to strengthening the paper.
>
> ---
>
> **Reference**
>
> [1] Nam et al., Spread Spurious Attribute: Improving Worst-Group Accuracy with Spurious Attribute Estimation, ICLR 2022
>
> [2] Ghosal and Li, Distributionally Robust Optimization with Probabilistic Group, AAAI 2023
>
> [3] Han and Zou, Improving Group Robustness on Spurious Correlation Requires Preciser Group Inference, ICML 2024

---

> > ### Comment · Reviewer_PVZF · 2025-08-04
> >
> > Thanks for the detailed response, it addressed my concerns. I will keep my positive score.

---

### Official Review · Reviewer_xJtj · 2025-07-01

**Clarity:** 3
**Significance:** 3
**Originality:** 3
**Rating:** 4
**Confidence:** 4

**Summary:**

This paper proposes a hierarchical extension to Group Distributionally Robust Optimization that addresses both inter-group and intra-group distribution shifts. The method improves robustness, especially for minority groups, by considering both group-level and within-group uncertainties. Experiments on datasets like CMNIST, Waterbirds, and CelebA show that the method outperforms existing techniques in handling distribution shifts.

**Questions:**

1. Could the authors provide a more detailed definition and explanation of "spurious correlation"?
2. What is the computational cost of the proposed method?
3. Are there any statistical guarantees associated with the method, especially regarding its performance in terms of mitigating spurious correlations?

**Ethical Concerns:**

["NO or VERY MINOR ethics concerns only"]

**Final Justification:**

I believe the paper's originality and theoretical depth are not sufficient to warrant a score of 5. Therefore, I keep my original score.

**Limitations:**

Yes.

**Paper Formatting Concerns:**

None.

**Quality:**

3

**Strengths And Weaknesses:**

Strength: This paper introduces a new method that improves robustness for both inter-group and intra-group distribution shifts. It is supported by both theoretical foundations and strong experimental validation.

Weakness: The hierarchical approach may come with increased computational costs. A discussion of computational efficiency and the method’s performance on larger datasets would strengthen the practical applicability of the approach.

---

> ### Author Rebuttal · Authors · 2025-07-31
>
> Thank you for the helpful feedback. We address the reviewer’s main questions below.
>
> ---
>
> **(W1 \& Q2)**
> >The hierarchical approach may come with increased computational costs. A discussion of computational efficiency and the method’s performance on larger datasets would strengthen the practical applicability of the approach. What is the computational cost of the proposed method?
>
> Thank you for raising this important point. While our method introduces additional modeling flexibility—specifically, by defining ambiguity sets using Wasserstein distances in the latent space—it remains computationally efficient in practice.
>
> As described in Algorithm 1, we use a single-step gradient ascent to compute the inner maximization and apply perturbations in the latent space. This only requires backpropagation up to the final layer of the network, avoiding the need for full backpropagation through the encoder. Moreover, the latent space (e.g., $2048$ dimensions) is significantly lower-dimensional than the input space (e.g., $3 \times 224 \times 224$), which further reduces computational cost.
>
> As a result, our method incurs only about $1.3×$ the training time of standard GroupDRO, while providing substantially improved robustness to intra-group shifts. We agree that clarifying these practical considerations would strengthen the paper, and we will include a discussion of training time and implementation details in the revised version.
>
> Regarding scalability, we fully agree that evaluating the method on larger-scale datasets would further strengthen its practical relevance. We would like to note, however, that our current experiments build upon standard and widely adopted benchmarks in the spurious correlation literature, including CelebA, Waterbirds, and CMNIST. These datasets have been consistently used in prior works due to their clear group structures and well-defined spurious attributes, and are commonly viewed as representative of real-world robustness challenges in structured group settings.
>
> That said, we appreciate the reviewer’s suggestion and agree that applying our method to larger-scale datasets—such as iWildCam2020-WILDS, as also mentioned by Reviewer y2Kb—would be a valuable direction for future work. We believe our method’s efficient latent-space formulation makes it a promising candidate for such settings, and we plan to explore this in follow-up experiments.
>
> ---
>
> **Q1**
> >Could the authors provide a more detailed definition and explanation of "spurious correlation"?
>
>
> We appreciate the reviewer’s suggestion. In our current draft, we use the term “spurious correlation” to refer to statistical dependencies between input features and labels that are present in the training data but do not reflect causal or semantically meaningful relationships.
>
> For instance, in the Waterbirds dataset, the background is spuriously correlated with the label: most "landbirds" are photographed on land, while most "waterbirds" appear on water. A model trained without accounting for such spurious signals may rely on them for prediction, resulting in poor generalization when these correlations no longer hold in the test set.
>
> We agree that this definition should be made more explicit in the Preliminaries section. In the revised version, we will formally introduce the concept of spurious correlations, along with motivating examples that clarify its relevance in the context of group distribution shifts.
>
> ---
>
> **Q3**
> >Are there any statistical guarantees associated with the method, especially regarding its performance in terms of mitigating spurious correlations?
>
> Thank you for the thoughtful question. While our method does not come with new formal guarantees specifically tailored to spurious correlation mitigation, it is built upon the GroupDRO framework, which is already known to provide strong theoretical guarantees for worst-group risk minimization and robustness to spurious features.
>
> Our contribution extends this framework by introducing a hierarchical ambiguity set that models both inter-group and intra-group uncertainty, while preserving the convergence guarantees, as shown in Appendix B. This allows our method to maintain desirable robustness properties while addressing a richer class of distribution shifts.
>
> ---
>
> We sincerely appreciate the reviewer’s insightful suggestions, which we believe have meaningfully strengthened the paper in both clarity and scope.

---

> > ### Comment · Reviewer_xJtj · 2025-08-06
> >
> > Thank you for the response. While I appreciate the authors' efforts and clarifications, I believe the paper's originality and theoretical depth are not sufficient to warrant a score of 5. Therefore, I will keep my original score.

---

### Official Review · Reviewer_7umr · 2025-07-02

**Clarity:** 3
**Significance:** 2
**Originality:** 2
**Rating:** 3
**Confidence:** 4

**Summary:**

The paper tackles subpopulation shift by focusing on the more demanding setting of intra-group distribution shifts. It extends GroupDRO with a hierarchical Wasserstein ambiguity set, enlarging the uncertainty region allowed within each group. To evaluate the idea, the authors craft image benchmarks that introduce controlled distribution shifts inside a group (through rotations or selective attribute changes) between training and testing. The proposed method then delivers strong worst-group accuracy across various benchmarks, with especially large gains when inter-group shifts are severe.

**Questions:**

- In the shifted-image benchmarks, does the validation split contain the same perturbations used at test time (e.g., rotated images), those at training time?
- Could you clarify how you define “inter-group” versus “intra-group” in the preliminaries?
- How much additional wall-clock training time does your method incur compared with standard GroupDRO?

**Ethical Concerns:**

["NO or VERY MINOR ethics concerns only"]

**Final Justification:**

My concern about insufficient experimental validation persists. The work omits several standard benchmarks and demonstrates effectiveness only within the vision domain, overlooking the established practice of evaluating textual spurious correlations in the literature.

**Limitations:**

yes

**Quality:**

3

**Strengths And Weaknesses:**

Strength
- The introduction of the intra-group distribution shift is meaningful.
- The proposed method is intuitive and effective.
- The paper was well-written and easy to follow.

Weakness
- My biggest concern with this paper is its limited novelty compared to prior structured DRO papers. Hierarchical or multi-level Wasserstein ambiguity sets have appeared before in robust representation learning [1] and in Group-DRO style RL formulations [2]. The present work’s novelty is therefore incremental, where its main distinction is the subpopulation-wise robust learning in vision datasets. Algorithmically, it remains close to vanilla GroupDRO.
- Standard benchmarks usually include CivilComments-WILDS and MultiNLI; although I don’t expect results on their specially shifted text variants, the paper should still report performance on the original (unshifted) versions of these datasets.
- The selection of $\epsilon$ is very heuristic and costly.

[1] Hierarchically Robust Representation Learning. CVPR 2020

[2] Group Distributionally Robust Reinforcement Learning with Hierarchical Latent Variables. AISTATS 2023

---

> ### Author Rebuttal · Authors · 2025-07-31
>
> Thank you for your thoughtful feedback. We appreciate the opportunity to clarify our contributions.
>
> ---
>
> **W1**
> >My biggest concern with this paper is its limited novelty compared to prior structured DRO papers. Hierarchical or multi-level Wasserstein ambiguity sets have appeared before in robust representation learning [1] and in Group-DRO style RL formulations [2]. The present work’s novelty is therefore incremental, where its main distinction is the subpopulation-wise robust learning in vision datasets. Algorithmically, it remains close to vanilla GroupDRO.
>
> We thank the reviewer for pointing out the relevance of [1]. While both our work and [1] explore hierarchical frameworks for robustness, [1] focuses on improving performance transfer across tasks by modeling robustness at the example and concept levels. In contrast, our method differs fundamentally in both motivation and, particularly, in how the hierarchy is conceptualized and implemented. Below, we clarify these distinctions.
>
> Although [1] also uses the term "hierarchical", its formulation does not reflect a generative or structurally grounded hierarchy in the way ours does. Our formulation explicitly models group-level uncertainty (e.g., due to varying group proportions) and connects it to instance-level uncertainty within **each group**, resulting in a coherent hierarchical ambiguity structure.
>
> By comparison, [1] treats concept-level and example-level robustness as separate components and does not link them via group-specific uncertainty. In fact, example-level robustness is applied uniformly, regardless of concept structure. As a result, this robustness is **ultimately achieved through standard $\ell_2$ regularization** on model parameters $\theta$, as shown in Equation (4) of [1]—a conventional mechanism that differs fundamentally from our hierarchical DRO formulation. By contrast, our method assigns larger uncertainty to underrepresented or internally diverse groups, allowing the model to dynamically allocate robustness where it is most needed. This hierarchical ambiguity structure enables more principled and targeted regularization by reflecting the relative uncertainty across both group and instance levels—an important capability that is not explicitly present in [1]'s formulation. Also, as shown in Section 3.2 and Algorithm 1 of [1], their formulation defines perturbations in the input space, whereas our method operates in the latent space—enabling the model to better capture semantic variation and improving robustness to realistic distribution shifts.
>
> We also appreciate the reviewer’s mention of [2], which introduces a Hierarchical Latent MDP framework for group-robust reinforcement learning. While both works use the term “hierarchical”, the underlying applications and modeling contexts differ.
>
> In [2], the hierarchy emerges from **applying standard Group DRO** principles to reinforcement learning tasks, where the environment first samples a latent task mode $z \sim w(z)$, followed by a specific MDP $m \sim \mu_z(m)$. This setup can be viewed as a probabilistic grouping structure—$w(z)$ represents group proportions and $\mu_z(m)$ defines a group-conditional distribution—thus aligning with the conventional Group DRO formulation (in our notation, $\beta$
> for group proportions and $P_g$ for group-conditional distributions), where a group $g \sim \beta$ and data $(x, y) \sim P_g$. From this perspective, the hierarchical structure in [2] mirrors the **group-wise structure in standard Group DRO**, and does not introduce fundamentally new sources of uncertainty. In contrast, our work explicitly models instance-level uncertainty within each group, which is a key limitation of existing Group DRO methods that we aim to address.
>
> We thank the reviewer for pointing us to these related works. The comparison has helped us better clarify the novelty and positioning of our method. In particular, it highlights how our framework addresses a previously overlooked yet important aspect of spurious correlation—robustness under underrepresented or internally diverse groups—which we hope can offer a distinct and meaningful contribution to the broader DRO literature.
>
> ---
>
> **W2**
> >Standard benchmarks usually include CivilComments-WILDS and MultiNLI; although I don’t expect results on their specially shifted text variants, the paper should still report performance on the original (unshifted) versions of these datasets.
>
> We appreciate the reviewer’s suggestion. We agree that including results on widely used NLP benchmarks such as CivilComments-WILDS and MultiNLI would provide a broader perspective on the method’s applicability. We had also noted this as a future direction in our conclusion and had planned to expand our evaluation accordingly. We are currently setting up the experiments for MultiNLI and will share preliminary results during the rebuttal period if time permits. In line with the reviewer’s suggestion, we will incorporate these results into the revised version.
>
> ---
>
> **(W3 \& Q1)**
> >The selection of $\epsilon$ is very heuristic and costly.  In the shifted-image benchmarks, does the validation split contain the same perturbations used at test time (e.g., rotated images), those at training time?
>
> We’d like to address W3 and Q1 jointly, as the question regarding $\epsilon$ selection is closely related to our validation setup discussed in Q1.
>
> We acknowledge that the selection of $\epsilon$ involves heuristic tuning and may incur additional computational overhead. However, this reflects a deliberate design choice in our framework, which extends GroupDRO to explicitly model structured uncertainty. Naturally, such flexibility introduces overhead, but it also enables a more principled treatment of intra-group variation.
>
> As a clarification related to Q1, we emphasize that we did not include test-time perturbations (e.g., rotated images) in the validation set. Instead, we followed the original train/validation split provided by each benchmark to maintain fairness in evaluation and avoid tuning $\epsilon$ with access to test-time information. This decision helps ensure robustness to realistic, unseen intra-group shifts.
>
> We wish to point out that the need to tune robustness-related hyperparameters such as $\epsilon$ is not unique to our setting. It arises broadly in robust learning frameworks, including adversarial training and other DRO-based methods. In fact, it is not uncommon in these areas to select $\epsilon$ arbitrarily or to report performance over a range of values, as we do in Appendix G.2.
>
> In the context of domain adaptation, it is also often considered acceptable to use a small validation set that reflects the target distribution—analogous, in our case, to including test-time perturbations in the validation set. While this can simplify the tuning procedure, we deliberately avoided this strategy to preserve the integrity of our robustness evaluation.
>
> We also note that even in [1], which the reviewer kindly cited, the perturbation radius is selected from a specified range, but the precise details of the tuning procedure—such as whether a downstream validation set was used or how the final value was determined—are not fully described. Rather than viewing this as a limitation, we believe this reflects a broader challenge in robust learning: selecting robustness-related hyperparameters without direct access to the target distribution.
>
> In our work, we aimed to provide a simple and effective guideline for selecting $\epsilon$, without relying on target-specific information, in hopes of contributing a more practical and transparent approach to robust model selection.
>
> ---
>
> **Q2**
> >Could you clarify how you define “inter-group” versus “intra-group” in the preliminaries?
>
> We thank the reviewer for pointing this out. In our current draft, we define “inter-group” shifts as distributional changes in the group proportions across training and test domains, which has traditionally been the main focus of GroupDRO methods. In contrast, “intra-group” shifts refer to changes in the conditional distribution $P((x,y) \mid g)$ within each group—e.g., shifts in data generation processes or subpopulation characteristics that affect how examples are distributed within a given group in practice.
>
> We agree that these definitions should be made explicit in the Preliminaries section for clarity. We will revise the paper to formally introduce and distinguish between inter-group and intra-group distribution shifts at the beginning of Section 3.
>
> ---
>
> **Q3**
> >How much additional wall-clock training time does your method incur compared with standard GroupDRO?
>
> Thank you for the question. While our method introduces additional modeling flexibility—specifically, by defining ambiguity sets using Wasserstein distances computed in latent space—it remains computationally efficient in practice.
>
> As shown in Algorithm 1, we use a single-step gradient ascent to compute the inner maximization and apply perturbations in the latent space. This only requires backpropagation up to the final fully connected layer of the network, avoiding the need for full backpropagation through the encoder. Moreover, the latent space (e.g., $2048$ dimensions) is significantly lower-dimensional than the input space (e.g., $3 \times 224 \times 224$), which further reduces computational cost.
>
> As a result, our method incurs only about $1.3×$ the training time of standard GroupDRO, while offering significantly improved robustness to intra-group shifts. We also agree that clarifying these practical considerations would benefit the paper, and we will include a discussion of training time and implementation details in the revised version.
>
> ---
>
> We sincerely thank the reviewer for the detailed and constructive feedback, which helped us refine both the clarity and scope of our contributions.

---

> > ### Comment · Reviewer_7umr · 2025-08-05
> >
> > Thank you for the thoughtful rebuttal. While I appreciate the clarifications, additional datasets remain necessary to validate the proposed method. To clarify: I do not suggest this work as a full replica of existing works. Rather, as it adapts established techniques to a new domain (which could be substantial contributions), it requires more robust experimental validation than currently provided. The present evaluation scope is insufficient to demonstrate the method's effectiveness in this novel context. I share similar concerns with Reviewer drEy.

---

> ### Author Response · Authors · 2025-08-06
>
> Thank you again for your thoughtful comments and engagement. We sincerely appreciate the opportunity to address your concerns and further clarify the intent and contributions of our work.
>
> ---
>
> **1. On the Scope and Sufficiency of Empirical Validation**
>
> We appreciate the emphasis on rigorous empirical validation. At the same time, we would like to clarify that the scale and structure of our experimental evaluation are consistent with established standards in this research area.
>
> Our work includes three widely used vision benchmarks (CMNIST, Waterbirds, CelebA), each exhibiting distinct types of spurious correlations, along with three corresponding variants that introduce carefully controlled intra-group distribution shifts. This results in six distinct evaluation scenarios, all motivated by realistic and principled failure modes. This level of coverage is comparable to many recent influential works: for instance, [1], one of our main baselines, evaluates on three datasets, and other recent studies on group robustness and spurious correlations typically use 3–4 datasets [2, 3, 4, 5]. Furthermore, a very recent paper that, like ours, investigates a novel failure mode within the spurious correlation setting—specifically, when spurious signal strength is low —reports accuracy results on only two original datasets [6].
>
> As suggested by the reviewer, we additionally conducted experiments on MultiNLI (unshifted) during the rebuttal period. These results support the practical relevance of our method, even on unmodified benchmark datasets.
>
> The results are based on three independent runs (seeds 0, 1, and 2). We followed the same hyperparameter settings as standard Group DRO implementations: batch size 32, a fixed linearly-decaying learning rate starting at 0.00002, the AdamW optimizer, dropout, and no $\ell_2$ penalty, as in prior work.
>
> Notably, our method outperforms Group DRO even though we did not introduce explicit distribution shifts within the minority group. We believe this performance gap arises from an implicit distribution shift: the number of minority group examples in the test set is noticeably smaller than in the training set (1521 vs. 886), increasing the likelihood of mismatch between the train and test distributions—precisely the kind of scenario our method is designed to handle more effectively.
>
> A similar pattern appears in CelebA (1387 in training vs. 180 in test), where our method shows a noticeable gain despite the absence of induced shifts. In contrast, in datasets like Waterbirds (56 vs. 642) and CMNIST (2998 vs. 8966), where the minority group is more prevalent at test time, Group DRO and our method show similar performance. These trends provide further support for the practical utility of modeling intra-group uncertainty, even in standard benchmark settings.
>
> |Method|Avg Acc| Worst Acc|
> |-|-|-|
> |Group DRO|79.6 ± 1.0|74.8 ± 0.5|
> |Ours|79.7 ± 0.9| 76.6 ± 1.3|
>
> ---
>
> **2. On the Realism and Relevance of the Evaluation Setting**
>
> With regard to concerns raised jointly with Reviewer drEy about real-world relevance, we would like to emphasize that our work addresses a practically important but underexplored phenomenon: when a minority group has too few samples to reliably reflect its underlying distribution, standard robust methods such as GroupDRO can fail—despite having access to group annotations.
> This setting is statistically well-grounded, as small sample sizes naturally lead to higher variance and less reliable estimates of the underlying distribution.
>
> Moreover, we would like to highlight that exploring new failure modes within the spurious correlation framework is an actively growing area of research. For instance, recent works have studied imperfect group partitions [7], multiple spurious features [8, 9], and low spurious signal strength [6]. These studies—like ours—highlight new challenges by modifying standard benchmark datasets rather than constructing entirely new ones. This reflects an accepted practice in the literature, which ensures comparability and reproducibility while exposing meaningful limitations in existing methods.
>
> In fact, we believe this represents a fair and realistic way to evaluate new ideas within the spurious correlation setting, precisely because it builds directly on widely adopted datasets. This perspective is supported by the design choices made in these studies and aligns with the broader goals of robustness research.
>
> We would also like to note that some of these studies (e.g., [6,9]) do not apply their modified settings across all datasets (e.g., no modified version for Waterbirds), whereas our evaluation remains more consistent and uniform across benchmarks. We believe this consistency further strengthens the empirical grounding of our study.
>
> We also note that, in line with Reviewer PVZF’s concerns about generalizability, we included controlled synthetic experiments, which are detailed in our response to their review for your reference.

---

> ### Author Response · Authors · 2025-08-06
>
> **3. On the Methodological Novelty**
>
> Finally, we would like to emphasize that our method is not a simple combination of existing components, but a principled extension of distributionally robust optimization to model multi-level uncertainty in a novel and tractable way.
>
> In our framework, introducing Wasserstein distance constraints gives rise to a multi-level ambiguity structure—an aspect that is uncommon in existing robust optimization formulations. We derive a tractable surrogate objective that bounds this formulation and provide convergence guarantees, enabling feasible optimization under structured distributional uncertainty.
>
> To further enhance practicality, we implement the training procedure in latent space, allowing the model to capture semantically meaningful shifts while significantly reducing computational overhead. In contrast to prior approaches, which generate augmented data in the input space and retrain the model in a separate step—resulting in a computationally expensive and less end-to-end pipeline—our method enables efficient, integrated training under multi-level uncertainty.
>
> ---
>
> We hope these clarifications address your concerns. We are sincerely grateful for your careful reading and suggestions, and we kindly ask that our evaluation strategy be interpreted in light of common practices around benchmark-based validation in this research area.
>
> ---
>
> **Reference**
>
> [1] Deng et al., Robust learning with progressive data expansion against spurious correlation, NeurIPS 2024.
>
> [2] Sagawa et al., Distributionally robust neural networks for group shifts: On the importance of regularization for worst-case generalization, ICLR 2020.
>
> [3] Nam et al., Spread Spurious Attribute: Improving Worst-Group Accuracy with Spurious Attribute Estimation, ICLR 2022.
>
> [4] Ghosal and Li, Distributionally Robust Optimization with Probabilistic Group, AAAI 2023.
>
> [5] Han and Zou, Improving Group Robustness on Spurious Correlation Requires Preciser Group Inference, ICML 2024.
>
> [6] Mulchandani and Kim, Severing Spurious Correlations with Data Pruning, ICLR 2025.
>
> [7] Zhou et al., Examining and Combating Spurious Features under Distribution Shift, ICML 2021.
>
> [8] Paranjape et al., AGRO: Adversarial Discovery of Error-prone groups for Robust Optimization, ICLR 2023.
>
> [9] Kim et al., Improving Robustness to Multiple Spurious Correlations by Multi-Objective Optimization, ICML 2024.

---

### Note · Authors · 2025-08-13

We sincerely thank the reviewers and the area chair for their time and thoughtful engagement throughout the review process. We sought to address all feedback through detailed clarifications.

We would like to take this opportunity to briefly reiterate the central contribution of our paper. Our work identifies a critical failure mode of existing methods for group robustness—particularly in the context of spurious correlations—where test distributions exhibit within-group distributional shifts not captured during training. Importantly, we demonstrate that this failure arises even in widely-used benchmarks simply by modifying the train–test split, without introducing any synthetic noise or external shifts. In such settings, existing methods degrade significantly and fail to maintain robustness.

To address this, we propose a method that models intra-group uncertainty via a hierarchical extension of the GroupDRO framework. In the rebuttal, we provided detailed clarifications regarding computational efficiency and hyperparameter tuning, which addressed earlier concerns and demonstrated the method’s practical feasibility. Furthermore, as we emphasized in our response to Reviewer PVZF, our method can serve as a drop-in replacement for GroupDRO in a wide range of recent pipelines. Since GroupDRO remains the backbone of many state-of-the-art approaches in this area, our method’s compatibility and improvements provide a practical path toward achieving comprehensive robustness in real-world distributional settings.

There was some concern regarding the real-world relevance of our setting. While we respect this perspective, we would like to clarify (as emphasized in our rebuttal, with references to several recent studies) that evaluating robustness through controlled modifications of standard benchmarks is a well-established and widely accepted practice. This approach ensures both reproducibility and comparability, which are core pillars of robustness research. Indeed, multiple reviewers explicitly recognized our benchmark design and problem setup as strengths of the work. We believe this disagreement stems more from differing interpretations of evaluation norms than from flaws in our methodology or experiments.

We truly appreciate the opportunity to share these final remarks and hope they help clarify the broader motivation and impact of our contribution, as well as encourage further work on this important yet underexplored challenge in group robustness research.

---

### Decision · Program_Chairs · 2025-09-17

**Decision:**

Reject

**Comment:**

The paper proposes a hierarchical extension of Group DRO in which the ambiguity set allows both shifts in group proportions and within-group distributional shifts (defined via a Wasserstein ball in the latent space). To solve this optimization problem, the authors propose to use an alternating optimization with PGD, mirror descent, and SGD steps. The authors test their method on modified versions of common subpopulation shift benchmarks, finding improvements in WGA.

**Strengths**
1. Targets an underexplored but important failure mode (PVZF, y2Kb)
2. Paper is well-written and easy to follow (PVZF, 7umr)
3. Consistent gains in worst-group accuracy on the shifted benchmarks and competitive results on originals (y2Kb, PVZF)

**Initial Weaknesses**
1. Significant performance improvements only arise on author-devised shifts, limited coverage of standard text benchmarks and larger-scale WILDS tasks (7umr, drEy, y2Kb).
2. Theoretical framing is close to prior hierarchical/structured DRO ideas, contribution may be incremental (7umr)
3. Extra hyperparameters and heuristic selection of $\epsilon$; potential runtime concerns (drEy, 7umr).

**Rebuttal Period**

The authors provided various clarifications, showed their runtime is only slightly worse than GroupDRO, and added a small unshifted MultiNLI result and a toy Gaussian example. Despite these efforts, 7umr and drEy maintained concerns about limited novelty and insufficient real-world evaluations.

**Overall Evaluation**

Almost all reviewers raised the issue of insufficient evaluation on real-world benchmarks (7umr, drEy, y2Kb), or the author-devised shifts being oversimplifed (PVZF). Thus, this should be resolved before the paper is ready for publication, but reviewers did not feel like this was sufficiently resolved by the author's rebuttals. During the private discussion period, neither of the "Accept" reviewers were willing to advocate strongly for acceptance of the work. For these reasons, I recommend rejection.